



# Statistical Characteristics of Mudflows in the Piedmont Areas of Uzbekistan and the Role of Synoptic Processes for their Formation

Gavkhar Mamadjanova[1,2], Simon Wild[1,3], Michael A Walz[1] and Gregor C Leckebusch[1]

[1]School of Geography, Earth and Environmental Sciences, University of Birmingham, UK
[2]Uzhydromet (Centre of Hydrometeorological Service of the Republic of Uzbekistan), Tashkent, Uzbekistan
[3]Barcelona Supercomputing Center, Barcelona, Spain

*Correspondence to*: Gavkhar Mamadjanova (gxm423@bham.ac.uk)

**Abstract.** The purpose of this study is to understand atmospheric factors from local to synoptic scales, which cause mudflow

variability on interannual and longer time scales. In a first step, historical data of mudflow occurrences in Uzbekistan provided by the Centre of Hydrometeorological Service of the Republic of Uzbekistan (Uzhydromet) for more than 140 years was analysed. During the investigation period a total of about 3000 mudflow events were observed with about 21 events per year on average. The majority of mudflows occur during the advection of westerly airflow when moist air from Central and Southern Europe reaches Uzbekistan. This synoptic weather type (SWT) can be related to one of the 15 primary synoptic circulation

types over Central Asia (CA) and Uzbekistan, which were subjectively derived by Bugayev and Giorgio in the 1930-40s. To understand the main atmospheric regimes steering the variability of mudflow occurrences, we applied additionally an objective classification following the Circulation Weather Types (CWT) approach. By means of the CWT approach, we further analyse that on mudflow-days the frequencies of cyclonic (C), westerly (W), south-westerly (SW) and north-westerly (NW) stream flows are increased in comparison to the climatological frequency of the occurrence of these circulation weather patterns.

Results confirm that CWT westerly airflow initiates relatively more mudflow events comparing to other CWTs in study area. Integrated approach of the CWT classification and an antecedent daily rainfall model are combined together in logistic regression analysis to construct mudflow triggering precipitation threshold per CWT class. In general W, SW and C weather types require less antecedent rainfall amount to trigger mudflow occurrences in the study area. This technique is thus shown to be applicable to coarse resolution climate model diagnostics.

**Key words:** mudflows, Bugayev's weather classification, circulation weather types, antecedent rainfall, logistic regression, Uzbekistan, Central Asia



# 1 Introduction

Mudflows are among the most damaging and deadly natural hazards in Uzbekistan. Data from the Centre of Hydrometeorological Service of the Republic of Uzbekistan (Uzhydromet) suggests that mudflows were responsible for over 38 deaths and damaged approximately 3000 households and 5000 ha of the agricultural crop over the past decade (2005–2014)

in Uzbekistan (Table 1). However, the incidence of damage may be much larger as these events commonly occur in mountainous areas, in incised valleys and in areas of otherwise low relief.

Hungr et al. (2014) suggest the term mudflow, as a very rapid, sometimes extremely rapid, surging flow of saturated plastic soil in a steep channel involving significantly greater water content relative to the source material. In the river basins of Uzbekistan, mudflows generally occur during the periods of intense rainfall or rapid snowmelt. The consistency of the mudflow

is mainly water and mud (liquidity index >0.5, e.g. Hungr et al. (2001)) with loose rock and other fragments, which flows down the hills and through the mountain streams. The destructive power of a mudflow can be greatly increased moving downhill by accumulating water and rocky mud. It can destroy riverbeds and banks of rivers, floodplains and even low terraces above the floodplain and other objects in its path (Chub et al., 2007).

The period of historically documented mudflow events confirms that the areas with a high passage of mudflow occurrences in

Uzbekistan can be divided into five regions: Fergana Valley, the Zerafshan Basin included in Zerafshan Valley, the Surkhandarya, the Kashkadarya and the Chirchik-Akhangaran rivers' basins (Figure 1). Precipitation is an important mudflow trigger (Huggel et al., 2012) in Uzbekistan; however, in mountain regions with snow cover and glaciers, slope instability and temperature (Huggel et al., 2010) are additional factors. Other main factors such as antecedent rainfall (Glade et al., 2000, Sidle and Ochiai, 2006) and rapid snowmelt (Kim et al., 2004) may further reduce the slope stability, thereby enhancing

potential of mud and debris flow occurrences.

Evidently rainfall records in high mountain regions are limited, reflecting the limited number of meteorological stations in Uzbekistan; thus the rainfall and mudflow data suffer due to missing spatial and temporal information. Furthermore, orographic effects on precipitation may not be captured adequately by a single (or a few) rain gauges (Huggel et al., 2012). In the Alps precipitation is generally observed in mid- to high-elevation areas (Buzzi et al., 1998, Huggel et al., 2012), similar to the

25 mountain ranges (Tien-Shan, Alai) in Central Asia (CA); however, mudflows usually occur in lower to intermediate elevations. Reviews on precipitation thresholds triggering landslides indicate the magnitude of an extreme event (Glade, 1998, Guzzetti et al., 2008) depends on the rainfall intensity and duration (Caine, 1980), the local climate and orographic precipitation (Buzzi et al., 1998, Gheusi and Davies, 2004), the geomorphologic structure of the area (Rosi et al., 2016), soil characteristics (Yamao et al., 2016) and land use (Sidle and Ochiai, 2006, Gravina et al., 2017).

Seasonal variations in precipitation and an earlier snowmelt are assumed to be the main factor for changes in the seasonality of mudflow occurrences in Uzbekistan. On this basis, the purpose of this paper is:

    a) to assess the link between the potential effects of synoptic conditions and the occurrence of extreme hydro-meteorological mudflow episodes in Uzbekistan;



b)  to establish an objective method for airflow classification of synoptic conditions which will be applicable to a large set of AOGCM members to investigate key factors of climate change impact on precipitation over Uzbekistan and CA;

c)  to compare and validate this with pre-existing subjective classification approaches;

5   d)  to identify thresholds of mudflow triggers taking into account the antecedent precipitation;

e)  and to validate the streamflow dependencies of these thresholds.

Section 2 describes available climatological data as well as the methodology contents of this paper. The main results presented in section 3 focuses on climatological and statistical characteristics of mudflow occurrences recorded more than 140 years in Uzbekistan thus making it accessible for an international scientific community for the first time. As an analysis of mudflow

conditions in Uzbekistan is widely missing in scientific, peer-reviewed international literature, this paper starts with a respective overview of local observational data and peer-reviewed material from non-English literature. Further we introduce a subjective method to classify synoptic conditions of CA and Uzbekistan as well as an objective approach known as Circulation Weather Types (CWT) to identify the major weather types leading to the formation of mudflows in study area in section 4. Precipitation thresholds triggering mudflow occurrences in Uzbekistan are quantified and discussed in section 5.

The main conclusion and discussion are summarised in section 6.

## 2  Research Approach, Data and Methods

The research approach consists of a five step strategy which is described in Fig. 2. The first component involves examining the historical data of mudflow occurrences in Uzbekistan and its characteristics on a longer time scale. Secondly, empirically developed local synoptic weather types (SWT) are manually assigned to the observed mudflow occurrences. The objective

CWT approach is therefore used to identify the atmospheric circulation and its relationship with the observed precipitation and their joint impact on mudflow occurrences in the study area. The fourth step is to evaluate the precipitation threshold triggering mudflow events in Uzbekistan using an empirical-statistical antecedent daily rainfall model (ADRM) (Glade et al., 2000). A detailed description of this approach as well as the datasets used for this component is given in the methods section and chapter 5 of this paper. Finally, the objective CWT method and the statistical model of ADRM are combined to estimate weather types

which are most likely to trigger mudflow occurrences in the study area.

The desired outcome of this study is to eventually select representative weather types which can then be applied to AOGCM and RCM. That way the influence of precipitation patterns on mudflow occurrences can be studied under climate change scenarios across Uzbekistan and CA in further studies.

### 2.1  Data

The investigation is based upon two categories of datasets: ground observation and reanalysis. Observed daily meteorological variables recorded by Uzhydromet, such as precipitation and temperature from meteorological stations located in the mountains and foothill zones with high mudflow passages were used to produce respective climatologies. In addition, historical data of





Uzhydromet regarding mudflow occurrences in Uzbekistan over the period 1870-2014 were analysed. It includes information such as the name of the water stream, location, date of passage, the potential reason for the formation of the mudflow and a rough estimate of the volume. Data of the daily synoptic situation or local classification of Synoptic Weather Types (SWT) for the years 1935-2014, which is available in Uzhydromet as a calendar of number coding of subjective types, was calculated to produce relative outputs.

In order to assess potential climatic drivers over Uzbekistan daily mean lower atmospheric flow in 700 hPa geopotential height (GPH) fields by ECMWF ERA-Interim reanalysis (Dee et al., 2011), spanning the period 1984-2013, was used to estimate the large-scale atmospheric circulation.

## 2.2 Methods

### 2.2.1 Circulation Weather Types (CWT)

The classification of daily flow patterns was done by the CWT approach. It was developed by Jenkinson and Collison (Jones et al., 1993) based on the original Lamb weather types for the British Isles (Lamb, 1972). The basic method and details of the scheme were provided by Jones et al. in 1993. The objective CWT scheme makes use of three basic variables that define the circulation features at the surface over the study region: direction of mean flow (D), the strength of mean flow (F) and the vorticity (Z). This basic method can be applied to any region with latitude ~30°–70° (Jones et al., 2013). For each day, the direction of the atmospheric flow is determined by considering pressure values at 16 points around the central point at 40°N - 67.5°E (Figure 10a), and is marked to one of ten CWTs: northeast (NE), east (E), southeast (SE), south (S), southwest (SW), west (W), northwest (NW), north (N), cyclonic (C), and anticyclonic (A).

Generally, the objective approach was successfully applied mainly in Europe (Trigo and DaCamara, 2000, Donat et al., 2010, Jones et al., 2013, Ramos et al., 2015), as well as other parts of the world, e.g. in Saudi Arabia (Kenawy et al., 2014). Recently, this method was used by Reyers et al. (2013) to determine present day and future high-resolution rainfall distributions in the Aksu river basin (on the southern slopes of the Tien-Shan Mountains, CA). To our knowledge this study is the first one that has applied the CWT approach for the territory of Uzbekistan. The aim of applying this method is to estimate the impact of airflow on precipitation patterns and its association with extreme mudflow episodes in Uzbekistan. Furthermore, the CWT scheme allows to define atmospheric flow regimes objectively to be applicable for climate model (AOGCM, CMIP5) data.

### 2.2.2 Application of Antecedent Daily Rainfall Model (ADRM)

In order to estimate the precipitation threshold causing mudflow events in the study area, a combination of an empirical and a statistical models is applied: 1) Antecedent Daily Rainfall Model (ADRM) as the relationship between antecedent rainfall conditions prior to an actual "rainstorm event" and the rainstorm magnitude itself (Glade et al., 2000); 2) a logistic regression model (LRM) as the relationship between an outcome (dependent or response) variable and a set of independent (predictor or explanatory) variables (Hosmer and Lemeshow, 2000).





The ADRM introduced by Crozier and Eyles (1980) defines landslide triggering rainfall conditions for the Ottago Peninsula during 1977-1978. This model was applied in many parts of the world (e.g. for New Zealand by Glade et al. (2000), for Portugal by Zêzere et al. (2005) and (2015), for Sao Miguel Island (Azores) by Marques et al. (2008), for Bangladesh by Khan et al. (2012), for China by Bai et al. (2014), etc.) to obtain the thresholds' probability of landslide occurrence events on the basis of

precipitation conditions. The advantage of this model lays in the substituting of soil moisture storage levels by daily precipitation data. In the absence of real-time soil moisture measurements this allows to predict the probability of landslides (Glade et al., 2000). The antecedent rainfall model (Crozier and Eyles, 1980) is expressed by the following formula:

$$ra_0 = kr_1 + k^2 r_2 + ... + k^n r_n \quad (1)$$

where $ra_0$ is the antecedent daily rainfall for day 0; $r_1$ is the rainfall on the day before day 0; $r_n$ is the rainfall on the n th day

before day 0; and $k$ is a constant <1.0. According to Davydov et al. (1973) and Bykov and Vasil'yev (1977) the $k$ parameter depends on the river morphometry and it varies from 0.7 to 0.9, and 0.5 in the case of a slack current. Crozier and Eyles (1980), following Bruce and Clark (1966), used a value of 0.84 for the k factor, which is close to that used in hydrological studies in North America, more precisely Ottawa (United States) streamflow data (Glade et al., 2000). Due to missing data on regional hydrograph recession curves in the study area, the constant decay factor, k in this research was assumed to be 0.84 based on

Crozier and Eyles (1980) and Crozier (1986). It was also suggested by specialists at Uzhydromet (personal communication) that $k \geq 0.8$ can be used for this study as it is used in the river catchments in the mountain areas of Uzbekistan. Setting $k$=0.84 worked satisfactorily in the ADRM to assess the triggering thresholds and mudflow probability in study area.

In the first step, daily rainfall totals of 30 years (1984-2013) recorded in four representative stations (Gallyaaral, Chimgan, Mingchukur and Sokh) were used to assess the average probability of mudflow triggering thresholds. The empirical model

analysis consists of calculating the antecedent precipitation for 10 consecutive days. Daily rainfall observations provided by Uzhydromet were used in this study and a period of 24 hr was taken between 8 a.m. of the previous day to 8 a.m. of the present day.

In the next step, LRM developed in order to estimate the relationship between mudflow occurrences and daily rainfall and antecedent rainfall value. The LRM was run in the freely available/open source R software environment. Mudflow occurrence

was treated as a dependent variable and 10 days of antecedent rainfall index and the rainfall value on the day with the mudflow event were used as independent variables. The values of the variables were the input data for the LRM algorithm to calculate the precipitation threshold equation and plot probability (P) curves for P=0.1, P=0.5 and P=0.9. This approach is analogous to the one presented in Glade et al. (2000).

Finally, we integrated the results from the two different investigation strands: applying a combination of CWT, ADRM

together with LRM to produce the precipitation threshold per CWT class and estimate of each weather type as a proxy for the triggering mudflows.





## 3 Climatic Conditions and Mudflows in Uzbekistan

### 3.1 General Climate Conditions

In general, the climate in Uzbekistan is continental and semi-arid with hot and dry summers and cold, sometimes severe winters with snowfall. Due to its geographic location (between 37°-45°N and 56°-73° E), Uzbekistan has three main climate zones: a

zone of deserts and dry steppes occupying about 79% of the territory, the foothills or piedmont zone, and the area of high mountains extending over the rest of 21% respectively (Chub, 2007).

The long-term climatology shows, that the mean air temperature in July varies from 26°C in a greater part of the lowlands to 30°C in the south and desert areas making it the hottest month of the year. The maximum values can reach up to 45°C in the southern part of Uzbekistan. The record temperature of 50°C occurred in Termez and the KyzylKum Desert. The coldest month

is January when the mean air temperature drops to 0°C in the south and -8°C in the north of the country. The minimum temperature can be well below -40°C in the Ustyurt Plateau in extremely cold years.

The air temperature in the piedmont areas at an altitude from 300-400 to 600-1000m is notably warmer during the cold season of the year and cooler in summers than the plain areas of the country. With increasing altitude in the mountainous regions the air temperature drops 0.6°C per 100 m on average. This is associated with the complex relief of the study area, e.g. cold air

flow over low-elevation, radiative cooling and wind speed (Bugayev, 1946, Chub, 2007, Kurbatkin, 2009). However, the temperature may be colder in the bottom of the valleys and intermountain basins due to frequent temperature inversions (Chub, 2007). Figure 3 characterises monthly mean temperature patterns and precipitation regime covering the 30-year period of January 1984–December 2013 inclusive, in four representative stations namely: Gallyaral (574 m), Chimgan (1620 m), Mingchukur (2132 m) and Sokh (1200 m), located in piedmont and mountain areas in Uzbekistan.

The amount and distribution of precipitation, as well as its seasonal variability, greatly depends on the geographical location of the area, topographic features and the general characteristics of the atmospheric circulation. In fact, several authors have identified moist air from the Atlantic Ocean, the Mediterranean Sea and the Persian Gulf as the main large-scale or regional climate factor for the precipitation regime in the country (Bugayev, 1946, Small et al., 1999, Inagamova et al., 2002, Chub, 2007, Schiemann et al., 2008).

The spatial distribution of the average precipitation in Uzbekistan shows a sharp contrast between the plain and mountain areas (Figure 4). Mean annual precipitation in major parts of the plain area or deserts and dry steppes (Ustyurt Plateau, Kyzylkum Desert, Karshi, Dalverzin and Golodnaya steppes) is about 80-200 mm. However, precipitation can be significantly greater in some piedmont areas and the mountains, particularly in the north-east and the south-east of the country. In fact, precipitation in areas with an elevation between 600-1000 m or piedmont areas (Tian Shan and Gissar-Alay mountain ranges) can reach up

to 500 mm; above 1000 m elevation the annual totals may exceed 500 mm. In some hillsides, especially the western slopes of Tian Shan, it may even be greater than 2000 mm (Chub, 2007).

Generally, the precipitation regime in Uzbekistan reveals a seasonal character with wet conditions from October to May and a dry season with little or almost no rainfall during the summer (Figure 3). Heavy precipitation events frequently occur during



the rainy season, especially in March and April. August represents the driest month with the minimal amount of rainfall throughout the year.

## 3.2 Mudflows in Uzbekistan

According to Lyakhovskaya (1989) and Chub et al. (2007) the earliest mudflow event induced by snowmelt and avalanches in Akhangaran River Basin was recorded in March 1870. The archive data of mudflow occurrences in Uzbekistan has been collected since then. Chub et al. (2007) states that there are more than 450 streams located in the river basins in the mountain and foothill areas where these extreme mudflow events usually occur in Uzbekistan.

For this study, data on mudflow occurrences in Uzbekistan was collected and arranged for a 145 year period from 1870-2014. During the observation period more than 3000 mudflow occurrences have been identified in the river basins in the piedmont areas of Uzbekistan. Table 2 provides comprehensive information about monthly distribution of mudflow occurrences in five regions of the country. Records confirm that mudflow events often affected the Fergana Valley indicating the highest event frequency during the investigation period of 12 mudflow occurrences per year on average (c.f. Table 2). Salikhova (1975) and Lyakhovskaya (1989) interpreted that due to its topographic feature, the Fergana Valley was susceptible to constant mudflow passages. Geologically the uplands in the Fergana Valley are mostly covered with loess loam which weakly infiltrates the water and makes it receptive even for a small surface runoff to flush off the soil easily. This process contributes to the developing of soil erosion which ends with the formation of mudflow episodes almost every year from March till August in the valley.

In contrast to the Fergana Valley, the geological structure of the mountain areas for the following regions, namely, Zerafshan, Surkhandarya and Kashkadarya Basins is formed of effusive or volcanic rocks of the Palaeozoic age, which may cause more debris flows rather than mudflow events there. Geomorphologic factors can be useful to determine the general susceptibility of various lithologies to landsliding in specific region (Sidle and Ochiai, 2006).

### 3.2.1 Statistical Analysis on Mudflow Data

We investigated century-long time series of the annual distribution of mudflows to identify key factors contributing to extreme mudflow occurrences. Mudflows with various magnitudes developing on the slopes of the study area appear several times during the year with an average number of 21 extremes per year (Table 2). The highest number of events was observed in 1930 with 167 mudflows followed by 161 mudflows in 1931, 144 episodes in 1963 and 108 events in 2012 (Fig. 5). An interesting point is that the frequency of mudflow events features a periodicity of approximately 30-years, which repeated the highest peaks in the 1930s, 1960s and 1990s (Fig. 5). The last peak period of mudflow activity occurred in the 2010s. This signature of potential natural variability will pose an additional challenge for investigation of mudflow cycles and their variability, under the future climate change conditions.

Due to Chub et al. (2007), more than 90% of all recorded mudflows were associated with extreme precipitation events, hail and sleet whereas 6% of mudflow episodes were observed during intensive snowmelt events induced by respective temperature



and precipitation changes. Glaciers melting due to increasing temperatures and outbursts from mountain lakes or water reservoirs are assumed to trigger further minor mudflows (1.4%) in the study area. Approximately 80% of all recorded mudflow episodes with different origins occurred during the period of April-June (Fig 6).

It is fairly common in Uzbekistan to observe mudflow occurrences in multiple streams on the same day. For instance,
Lyakhovskaya (1989) reports that on 15 April 1964 only in the Samarkand province (Zerafshan Valley) 22 mudflow episodes were recorded in a day. During the spanning period of 1870-2014 up to 24 events or the maximum passage of flows were observed on 18 May 1991 in many parts of the country.

## 4  Atmospheric Circulation over Central Asia and Uzbekistan

### 4.1  Synoptic Weather Types (SWT) and Mudflows

Early investigation of atmospheric circulation over Central Asia were started in 1921 in the Turkestan Synoptic-meteorological Institute in Tashkent (Aksarin and Inagamova, 1993). The complexity and diversity of the regional atmospheric circulation was identified soon after Bugayev and Giorgio (Giorgio and Bugayev, 1936, Bugayev et al., 1957, Aksarin and Inagamova, 1993) developed and simplified a model of airflow advection to explain for synoptic differences in the 1930s and 40s. The founders of the Central Asian Tashkent Institute of Weather Forecasters, Bugayev and Giorgio had supervised research on
synoptic meteorology and the impact of orographic factors on CA's climate for many years. In 1947 scientists had published the first findings of the statistical characteristics of synoptic situations over the region for the cold period in the USSR Academy of Science newsletters (Sarimsakov et al., 1947). After a decade, researchers had summarised the studies on the atmospheric circulation classification scheme for CA and had published it as a fundamental monograph, describing the main atmospheric patterns as 11 SWT over CA (Bugayev et al., 1957). This monograph is still being used as the main literature study and
guidelines on synoptic conditions in Central Asian countries mainly in Uzbekistan (personal experience, Aizen et al. (2004)). Figure 7 provides a basic illustrative view of SWT classified by Bugayev et al. (1957) emphasizing the air mass source regions and their path to CA. In the early 1960s, researchers at the Scientific Institute of Hydrometeorological Service of Uzbek SSR (nowadays Uzbekistan) updated Bugayev and Giorgio's classification from 11 up to 15 types by including additional weather classes. Table 1 (c.f. appendix section) provides comprehensive information for these 15 primary SWT, describing weather
conditions on a synoptic scale in CA and particularly in Uzbekistan. Figure 8 shows the seasonal distribution of SWT between 1935 and 2014. Daily data on the frequency of SWTs observed in Uzbekistan since 1935 is available as a hard copy in Uzhydromet Library Services and Archive Department.

During the investigation period between 1984 and 2013, there were more than 300 days with mudflow occurrences in Uzbekistan. Figure 9 shows the frequencies of mudflow days for each synoptic weather type in five regions (Zerafshan,
Fergana, Chirchik-Akhangaran, Kashkadarya and Surkhandarya) with high mudflow passage. According to the results, the majority of mudflows occur during the advection of airflow from *west* (SWT 10) and *low level of small barometric gradient* (SWT 13) in the study area. However, the SWT 13 is the most frequent weather type in Fergana Valley (Fig. 9b) as the





interaction of frontal circulation with orography as well as the associated effects of condensation and evaporation are assumed to determine the formation of low-level fronts and small-scale rainbands (Buzzi et al., 1998, Inagamova et al., 2002) there. *Stationary cyclone* type (SWT 8) is the second most frequent SWT triggering mudflow events although in some regions it is not as prominent. Purely anticyclonic weather types (SWT 9, 9a, 9b) constitute less than 15% of all events (Fig. 9) even though this weather type is the most frequent per year on average (Fig. 8). Cyclones propagating from the *south-west* (SWTs 1 and 2) towards the study area, *advection from north and north-west* (SWTs 6 and 5), *high level of small barometric gradient* (SWT 12) and synoptic wave activity on a cold front (SWT 7) contribute also to significantly unstable weather conditions inducing mud and debris flows (Fig. 9). While the majority of SWTs were associated with regional or local scale precipitation patterns, the summer thermal low (SWT 11) does not (Fig. 9 b, c, d). Observations confirmed this SWT has triggered mudflows with origins of snow and glaciers melting due to increasing surface temperatures.

## 4.2 Circulation Weather Type (CWT) Approach

### 4.2.1. CWT and Precipitation Climatology

In order to assess the impact of each CWT class on mudflow triggering precipitation regimes, long-term daily circulation types and the corresponding daily values of precipitation were analysed.

Figures 11-12 (c.f. Figures 2-3 in appendices) show the seasonal distribution of the relative frequency of the number of days and precipitation values for each CWT class during 1984–2013 for the four stations in total. The column graphs (a, b) in Figures 11-12 show the frequency of each weather class (CWT days, %) as well as the relative contribution of each weather types to the total recorded rainfall values (total precipitation, %) and the average daily precipitation per CWTs (mm/day). The box plots (c, d) in Figures 11-12 highlight the seasonal rainfall statistics including 90th and 95th percentile of precipitation value within each CWT class.

It is worth noting that on average the large-scale atmospheric circulation over Uzbekistan and Central Asia is mainly dominated by **westerly** (W) weather type throughout the year (Figures 11-12 a, b). The seasonal frequencies of W type shows the highest value between 22-38%, depending on the season. The percentage of precipitation during the W days ranges from 35% to 75% of the total annual precipitation per station, respectively. The spatial distribution of daily average precipitation up to 3-7 mm/day was associated with the W flow at nearly in all the stations (9-12 mm in Chimgan, Figure 2 a, b in appendix).

**Cyclonic** (C) and **anticyclonic** (AC) weather types feature almost similar frequencies (18%) in summer, however, C flow contributes roughly four times more of the annual precipitation (up to 27%) and daily rainfall values (13 mm/d) compared to the AC class. Even though AC circulation has higher reputation of the circulation days in contrast to C type in the winter half year, the C days still indicate higher values of precipitation pattern which makes this CWT one of the wettest airflow. The 90th and 95th percentiles of the precipitation totals can also confirm this (Figures 11-12 c, d).

**South-westerly** (SW) flow occurs from 6% to 13% during the year and contributes 10-22% of the precipitation totals (5% in Fergana Valley). Seasonal distribution of weather types associated with **easterly** and **southerly** flows (E, SE and S) are fairly





variable (up to 1.3 %) per season throughout the year and produce little or almost no rainfall, i.e. 0.2-0.5% or less than 1 mm/day. However, the box plot statistics indicate that S flow leads to precipitation values compared to the other directional classes within the southerly weather type (c.f. Fig. 3c in appendix). Another minor occurrences associated with **north-westerly** (NW) circulation (1-3%) contributing approximately 3% of the precipitation totals and 4 mm daily rainfall on average
depending on the area and the rain gauge data.

In comparison to other stations, the **north-easterly** (NE) and **northerly** (N) weather types have different precipitation patterns at Sokh station, Fergana Valley, during the warm season of the year (Fig. 12). This is due to the location and orographic pattern of the area (Fig. 10c) which makes these weather types one of the wettest airflow throughout the year in the area with frequent floods and mudflow occurrences.

The impact of small-scale orographic features on weather types and rainfall distribution is assumed to be one of the reasons for the notable seasonal variabilities of the **undefined** weather type throughout the year. For illustrative purposes, it was included ERA-Interim orography map which confirms this (Figure 10 c). An important inclusion in this study was the CWT evaluation which highlighted the spatial distribution of precipitation in Uzbekistan on synoptic scale.

### 4.2.2  CWT and Mudflows

The aim of this section is to analyse the relationship between CWT classes and mudflow occurrences in the investigation regions of Uzbekistan. Figure 13 allows a comparison between mudflow together with CWT daily frequencies in March-August, 1984-2013. For consistency within this study, the central grid point (40N-67.5E) was selected for investigation of 101 days induced mudflow occurrences in Zerafshan Basin, 147 days in Fergana Valley, 57 days in Chirchik-Akhangaran, 35 days in Kashkadarya and 44 days in Surkhandarya basins observed in a warm season during the period 1984-2013 (Figure 13). An
interesting point based on the analysis illustrated in Figure 13 is the similarity of the Zerafshan (a), Chirchik (c), Kashkadarya (d) and Surkhandarya (e) regions. However, complex orography and not simple topography affect the weather systems and precipitation distribution assumed to produce slightly different results for Fergana Valley (Fig 13b). The frequencies of W, SW, and C weather types are considered as a "rainy" group with increased mudflow episodes compared to the rest of the CWTs nearly in all five regions. There is also considerable amount of NW circulation type associated with extreme mudflow events
in Chirchik and Surkhandarya basins (Fig. 13 c, e). However, NE days trigger more mudflows in contrast to SW flow which reveals a decrease in Fergana Valley (Fig. 13 b).

Figure 14 confirms that the frequency of mudflow days per C, W, SW and NW classes is increased in comparison to normal CWT climatology and in contrast to the average number of mudflow days per CWT classification during a warm phase of 1984-2013. The AC, NE (except in the Fergana Valley), E, S and SE weather types have a noticeable decline of mudflow
frequencies compare to the climatological mean per CWT.

It is noteworthy here that the variability of C, W, SW and NW days in comparison to all CWT days again showing increased trends in mudflow probability and corroborated all results considered above. Therefore it can be concluded that appreciable



weather classes (C, W, SW, and NW) are the main contributors in agreement with the recorded precipitation distribution and observed mudflow occurrences for the study area.

## 5 Statistical Modelling of Precipitation Thresholds for Triggering Mudflows

### 5.1 Application of ADRM for Station Data

The model results (Figure 15) show that minimum and maximum thresholds boundaries with different probabilities of occurrence exist, but not all rainfall values between the thresholds are associated with mudflow episodes. A daily rainfall value ≥0.1 mm, mainly from March till August (1984-2013), was selected for plots presented in Figure 15. Following Glade et al. (2000), all rainfall days were divided into three categories: days associated with mud and debris flows; days with no recorded mudflows; and days with probable mudflow episodes. Events in the third category were not recorded in historical data. Thus,

it could not be assumed that there were no mudflows on those days as the rainfall value was similar or exceeded the precipitation level during the days of recorded debris and mudflows. However, the summary of the daily precipitation value, which is recorded from 8 am till 8 am of the next day might not tally with the day of the mudflow occurrences. Therefore, mudflow passages were also assumed to be a probable variable as we did not have the information regarding the exact time of the mudflow occurrence. Additionally, the air temperature patterns in March-May were analysed for some meteorological

stations; for example, Chimgan located at more than 1600 metres altitude, which led to the conclusion that the precipitation events recorded were likely snowfall periods which leaves very little possibility of mudflow passage. Hence, a probable mudslide event was not analysed for high altitude areas, even though high precipitation values existed.

According to Figure 15, a 10% probability of mudflow events may occur if the antecedent rainfall value reaches 40 mm in the Gallyaaral station, approximately 60 mm in Mingchukur and 90 mm in Chimgan stations (Table 3). Interestingly, there is

always a chance that a rainfall event of sufficient magnitude could induce the mudflows, even when the antecedent index is less than the levels above. The results indicate that a weather type with a high level of relative moisture may provide sufficient rainfall to trigger floods and mudslides even when the cumulative rainfall value is close to 0 or soil moisture storage is in deficit. In contrast, after a long period of accumulation of antecedent rainfall, which weakens the slope increasingly, the slope becomes more susceptible to the less value of rainfall on a given day.

Obviously some mudflow events were recorded when the rainfall level and antecedent rainfall index showed less than 10 mm. If the mudflow event was induced by the snowmelt due to a joint occurrence of temperatures and rainfall, then it is possible that less rainfall might be required to trigger the mudslides. On the other hand, locally heavy rainfall events in the adjacent areas of the stations could induce the flows and mudslides in the river catchment and hilly areas.

Figure 15 (d) shows that the Sokh station area located in the Fergana Valley is more susceptible to extreme mudflow events;

indicating that mudflow events are also influenced by the geomorphologic structure of the area. The 0.1 probability threshold indicates that 10 mm of rainfall with antecedent conditions of less than 30 mm can trigger flash floods or mudflows in Sokh.



It means that the threshold varies in space and it is important to consider the regional characteristics of the research area whilst applying the ADRM.

Table 4 provides logistic regression equations for the four stations' results respectively which can be used to estimate the rainfall thresholds with different probabilities of mudflow occurrences. For Chimgan station a cubic regression and for

Mingchukur a quadratic equation with probability curves proved to be the best fit, however, probability envelopes of the linear regression worked satisfactorily for the data of the other stations, namely Gallyaaral and Sokh. Associated values of Chi-squared test represented in Table 4 show the results of less than 0.001, which can confirm the significance of model fitting for the stations data.

## 5.2  Application of ADRM per CWT

In this section, the ADRM fit for each CWT class is examined in order to identify precipitation thresholds triggering mudflow events under each weather type for the four stations (Gallyaral, Chimgan, Mingchukur and Sokh) located in areas with high probability of mudflow events in Uzbekistan. For this purpose, all rainfall days with an amount of ≥0.1 mm and the calibrated antecedent rainfall was divided into each weather type. Mudflow events were marked for the weather types in which they were observed respectively. This investigation was evaluated for the summer periods (March-August) of 1984-2013 by using CWT

and ADRM together with LRM to construct mudflow triggering precipitation threshold per CWT class (Figures 16-17; c.f. Figures 4-5 in appendices).

The probability envelopes on C, W and SW days in all stations show consistently positive results each resembling regional overlaid threshold to trigger the extreme mudflow event. Following the above named airflows, the antecedent rainfall index associated with the AC circulation has a sufficient magnitude which could trigger mudflows even when the antecedent index

and the rainfall value are less than regional threshold for Gallyaral (Fig. 16), Chimgan (c.f. Fig. 4 in appendices) and Sokh stations (Fig. 17). It is assumed that AC hybrid, mainly anticyclonic westerly (ACW) and anticyclonic south-westerly (ACSW) initiate significantly more mudflow probability rather than purely anticyclonic flow. Another curious point of this approach is the overlaid threshold probabilities for mudflow events under the NW airflow in Chimgan, Sokh and Baysun (c.f. Fig. 5 in appendix), which indicate the similar or less values of antecedent and daily rainfall records of regional probability which can

trigger mudflow occurrence there. The NE (except in Fergana Valley, Fig 17), E, SE and S flows had little or no precipitation patterns that may affect to induce mudflow cases in the study area. Threshold probability tests computed for rainfall data per CWT for four individual stations over the period 1984 to 2013 are given in Table 2 in the appendices section.

This examination is an attempt to identify more sensitive weather types to trigger mudflow events in Uzbekistan, using the CWT approach and antecedent rainfall model, respectively. The relative importance of each CWT to induce mudflows varies

considerably, from the antecedent rainfall index to the daily precipitation value. Results from this study confirm that W, SW, C, NW and AC hybrid (associated with W and SW flow) are the main drivers of the interannual variability of precipitation patterns and responsible for the rainfall induced mudflow cases depending on the regions in Uzbekistan on synoptic scale. This



confirms core findings of the synoptic classification by Bugayev and the CWT objective approach can be seen as a useful extension to address e.g. questions of anthropogenic climate change with model data.

## 6 Summary and Discussion

Extreme precipitation events in Uzbekistan are responsible for about 90% of historical documented mudflows especially in the warm season (March-August) for the years 1870-2014. What are the main precipitation supporting weather types inducing mudflows in the study area? In the present study, mudflows and their relationship with precipitation and weather types were investigated using multiple and coherent systematic approaches within Uzbekistan. This is especially important as only few studies are investigating atmospheric circulation conditions and precipitation variability on different spatial scales over Uzbekistan. The principal findings of this study included previous investigations and qualitative comparisons are as follows:

1. Advection of moist and relatively cold air from the west, as classified in the SWT classification by Bugayev et al. (1957), was revealed as the dominant weather situation inducing mudflows. This result is consistent with the findings of Aizen et al. (2004), hereby confirming this synoptic pattern as a predominant synoptic scale driver for precipitation climatology in Central Asia.

2. The relationships and related variables explaining the spatial distribution of precipitation, obtained by an objective CWT approach, defined the four weather classes westerly (W), south-westerly (SW), cyclonic (C) and north-westerly (NW) as the main drivers of precipitation characteristics on a regional scale. This allows a positive evaluation of the CWT method in principle.

   Furthermore, CWT findings are in line with results from Reyers et al. (2013) who evaluated spatial patterns and annual cycles of precipitation using CWT scheme for Central Asia. Interestingly, the westerly airflow in Reyers et al. (2013) was split into two subgroups, as W1 (distinct zonal flow) and W2 (negative 700 hPa GPH gradient) and NE and E as well as SE and S were combined respectively. In general it was found that probability of precipitation was much higher for C, CWT W2, N and SW airflows during the summer. However, the highest rainfall probability and precipitation amount was attributed to the seldom CWT NE/E weather type (Reyers et al., 2013). In our study as well NE weather type despite its low frequency for the selected grid point revealed the high probability of precipitation patterns that could trigger mudflow events in Fergana Valley (Figures 12c, 13b, 14b and 17). Fergana Valley which has a better representation of topography presumably makes a case of particular interest with the findings in previous studies. According to Schiemann et al. (2008) it can be assumed that on smaller spatial scales, the influence of topography on precipitation climatology over Central Asia is paramount. Small et al. (1999) and Reyers et al. (2013) both confirm this.

3. By means of the antecedent rainfall model the regional different probabilities of precipitation thresholds causing mudflow events could be identified. However, sparse data on actual mudflows and uncertainty over probable mudflow occurrences could be the main factors regarding the uncertainty in model building (Glade et al., 2000)



4. A combination of three statistical approaches (CWT, ADRM and LRM) revealed that whilst the W, C and SW air flows occurred over Uzbekistan the antecedent rainfall affected to trigger mudflow episodes depending on the station data. Hence, above named flow directions bring more rainfall which can increase mudflow magnitude even though there is a small accumulated rainfall conditions.

Thus, it can be concluded that the CWT approach and ADRM produced robust results, despite the orographic influence on the study area and the limited data on mudflow timing and precipitation intensity. Future investigation should focus on regional downscaled seasonal and annual precipitation, with observed data per CWT, preferably W, C, SW and NW air flows, to identify key factors of future rainfall distributions and to discover how this will affect mudflow occurrences on a longer timescale.

**Data availability**

The ECMWF reanalysis data is available at http://apps.ecmwf.int/datasets/data/interim-full-daily/levtype=pl/

**Competing interests**

The authors declare that they have no conflict of interest.

**Acknowledgements**

We are grateful to Uzhydromet for sharing the mudflow and daily meteorological data. We are also grateful to ECMWF for
granting access to the ERA-Interim reanalysis data. G.M. expresses her gratitude to Islamic Development Bank (IDB) for awarding her with a PhD scholarship. Chapters 3 and 4.1 of this manuscript are the fragments from the MSc thesis of G.M. at the National University of Uzbekistan (NUUz). G.M. expresses her sincere gratitude to late Professor Gennady N Trofimov at the Department of Geography, NUUz, and to Professor Boris K Tsarev at Uzhydromet for their guidance and encouragement throughout and after her MSc programme. G.M. also sincerely appreciates the various researchers at Uzhydromet for their
timely inputs with regards to data clarifications. G.M.'s special thanks are extended to Nicolas Kirchner Bossi and Mohammad Alharbi (University of Birmingham) for their assistance with programming.

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



**Figures**

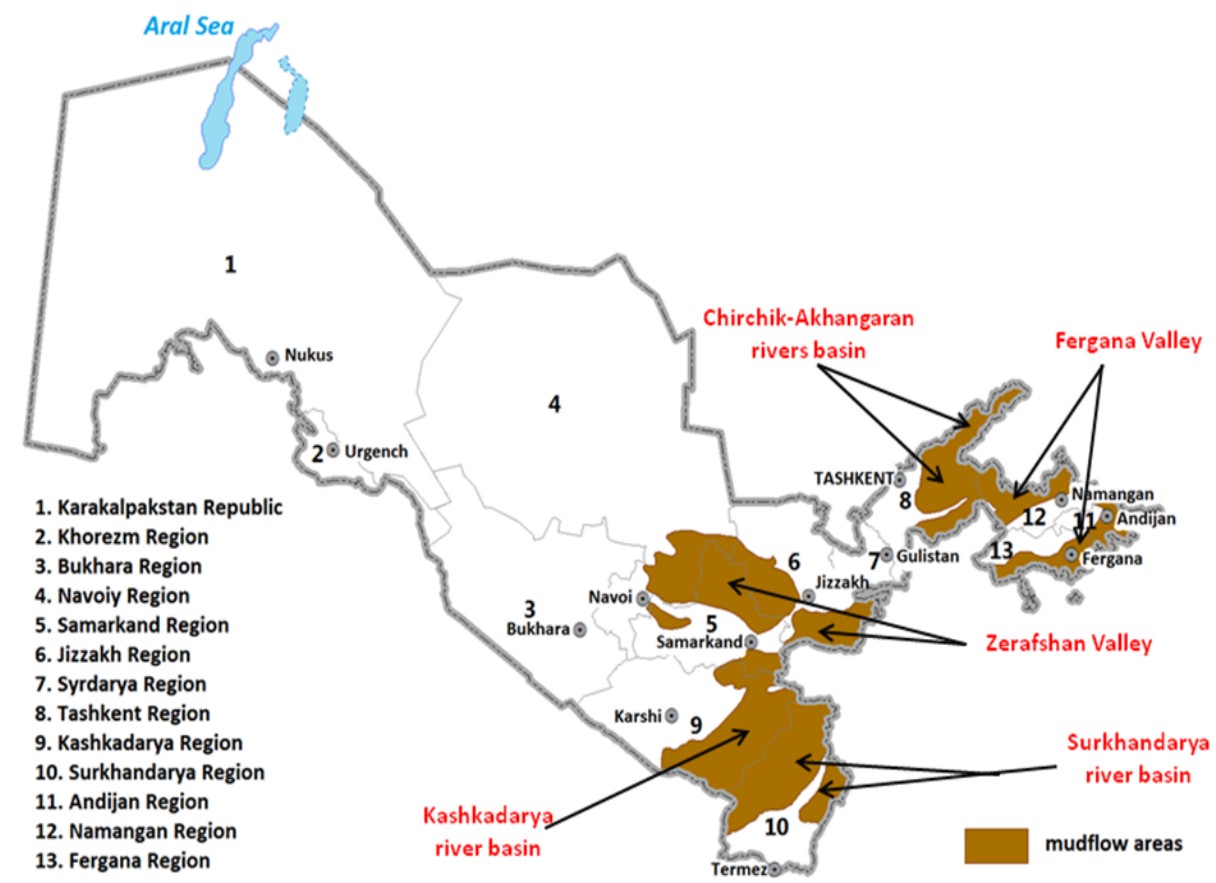

**Figure 1.** Study site in Uzbekistan including areas with high probability of mudflow passage: Fergana Valley in the east; Chirchik-Akhangaran Basin in the north-east; Zerafshan Basin in central part of the country; Surkhandarya and Kashkadarya rivers' basins in the south of Uzbekistan (Source: Uzhydromet).





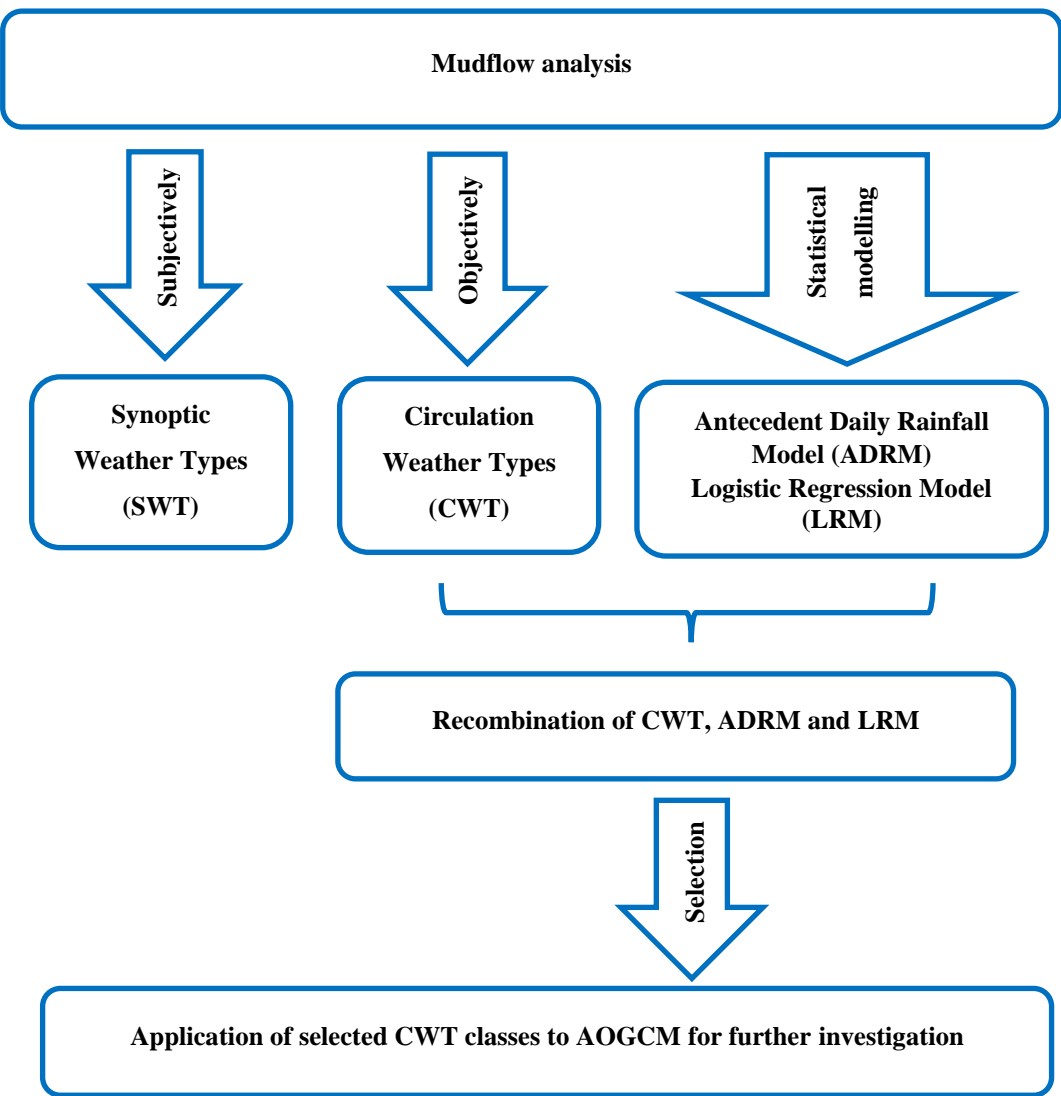

**Figure 2.** Schematic diagram of pathways by which the stages of investigation presented in this paper.



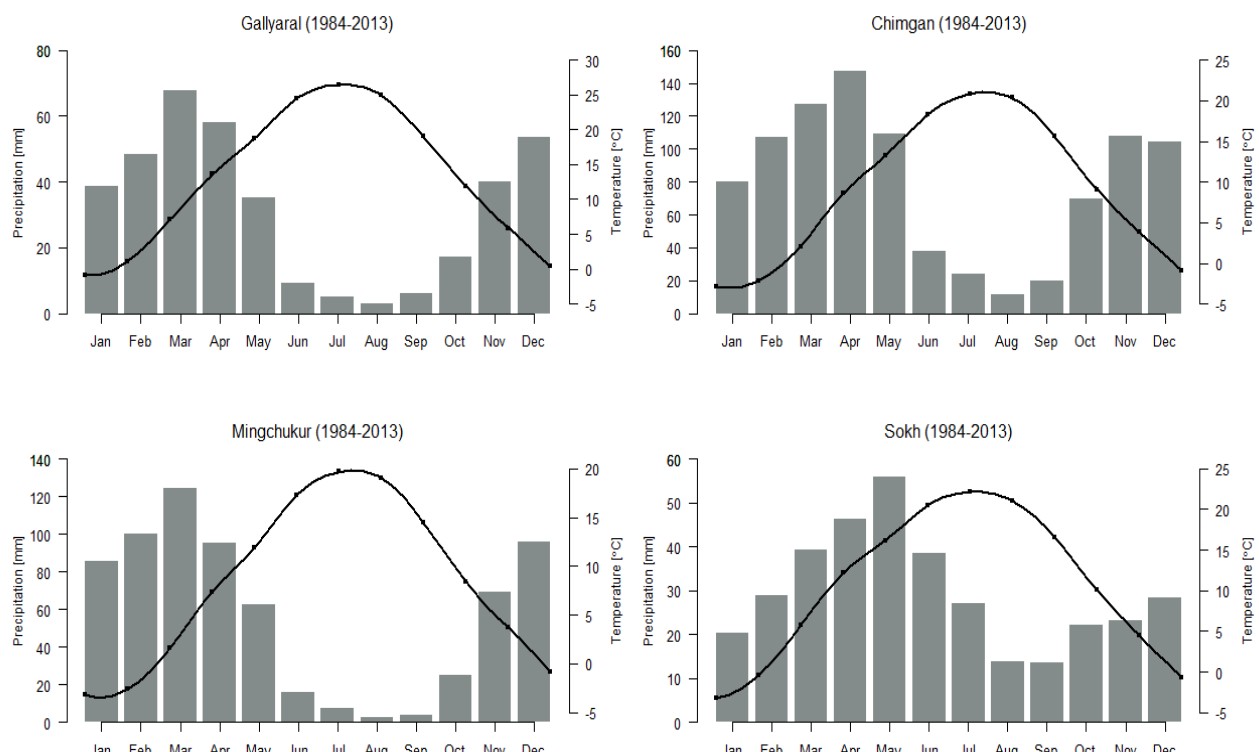

**Figure 3.** The 30-year means (1984-2013) of monthly temperature (˚C, black line) and precipitation (mm, grey bars) in four selected stations (Gallyaral, Chimgan, Mingchukur and Sokh) for each basin with high occurrences of mudflow in Uzbekistan. Graphs have different scales.





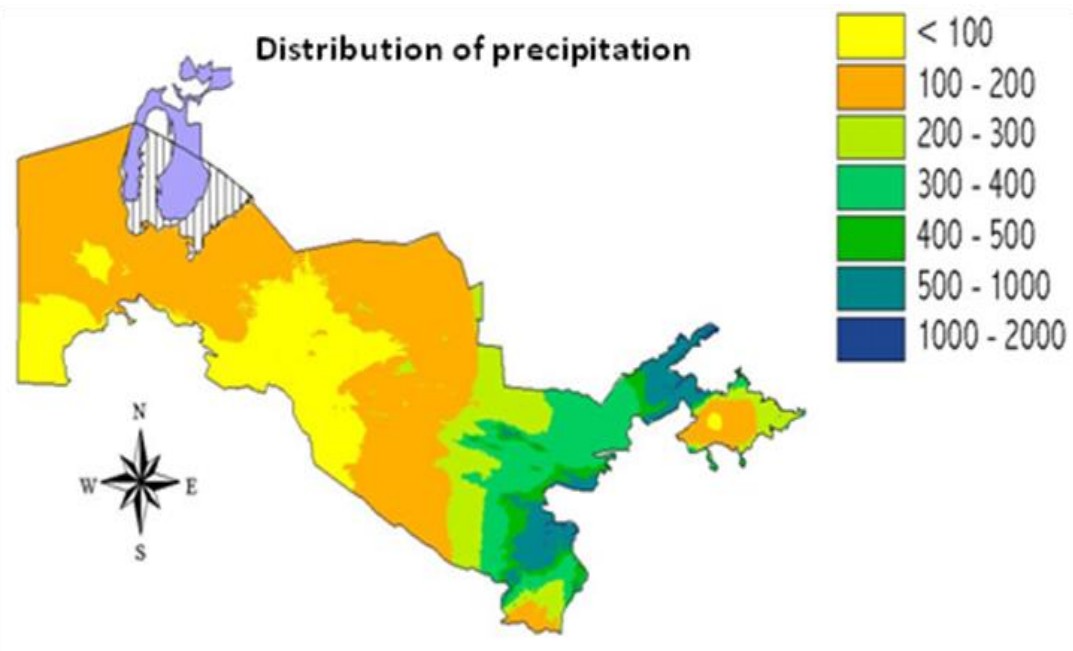

**Figure 4.** Map showing the spatial distribution of total annual precipitation (mm) in Uzbekistan for the 1961-1990 periods (Source: Uzhydromet).





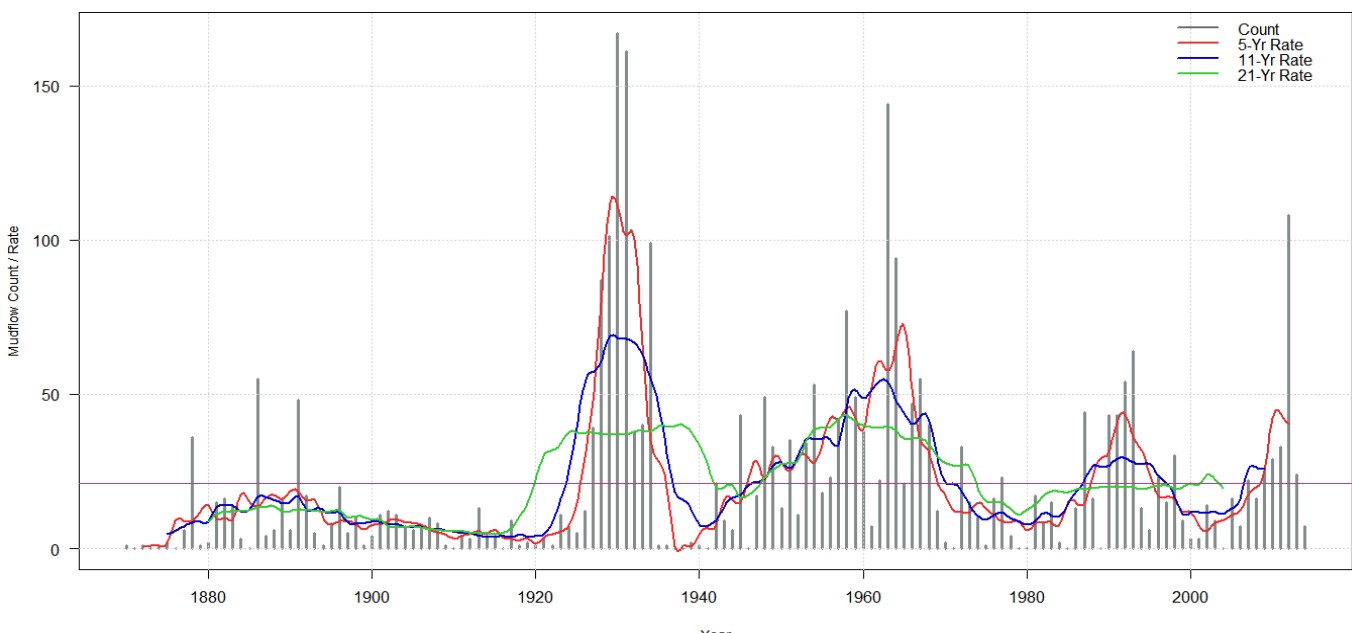

**Figure 5.** Variability of mudflow events in Uzbekistan (1870-2014). Vertical bars present the mudflow observations for each year. The mean annual mudflow count (21) is indicated in a solid continues horizontal line (pink). Curves (red, blue, green) have been fitted to the distribution for illustrative purposes, denote the 5, 11 and 21-year rates of mudflow occurrences.

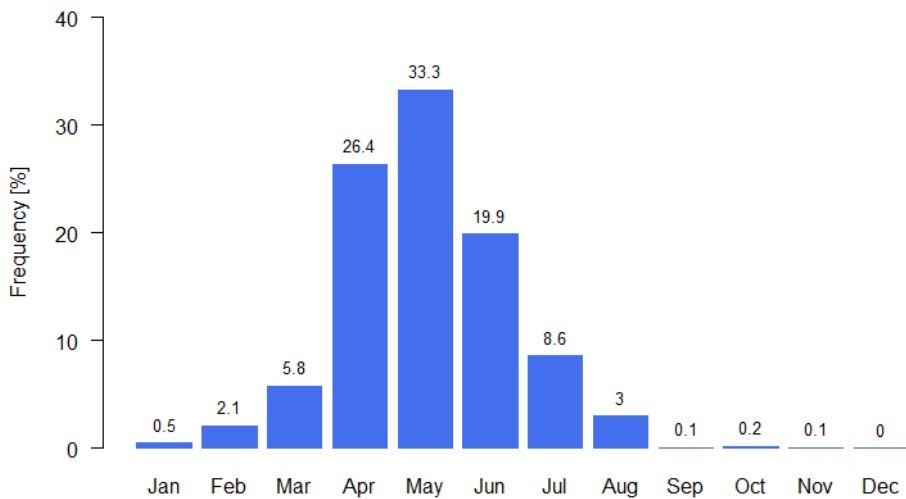

**Figure 6.** Distribution of monthly mudflow frequencies (bars) for the years 1870-2014. Values over the bars indicate the percentage of mudflow occurrences in a given month.




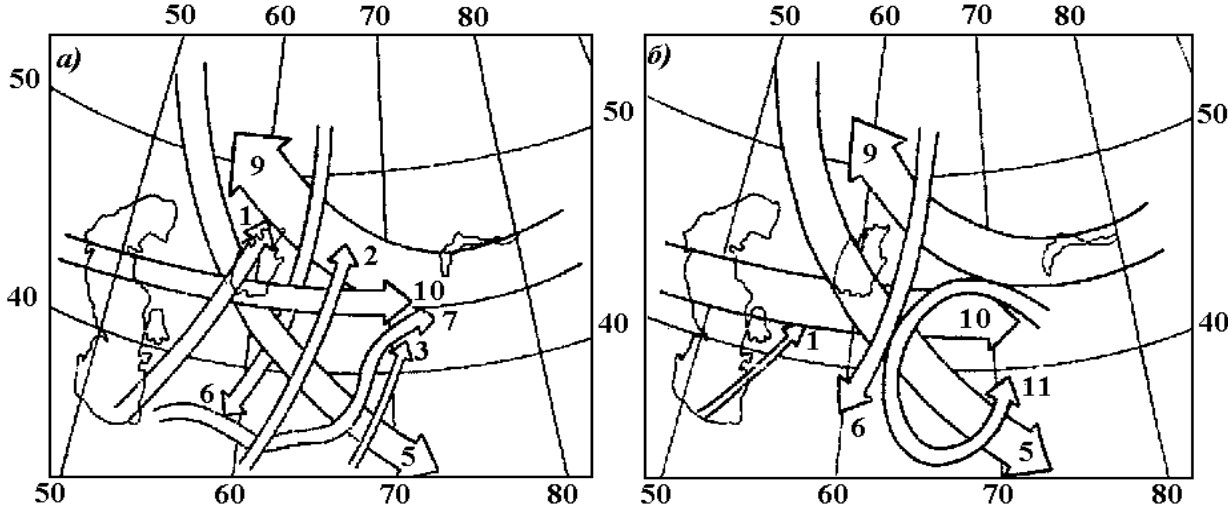

**Figure 7.** Scheme of synoptic weather types in Central Asia and Uzbekistan during the cold (a) and warm seasons (b) of the year. Numbers and cursors indicate the weather type and its approaching to the area of Central Asia and Uzbekistan (source: Inagamova et al., 2002).

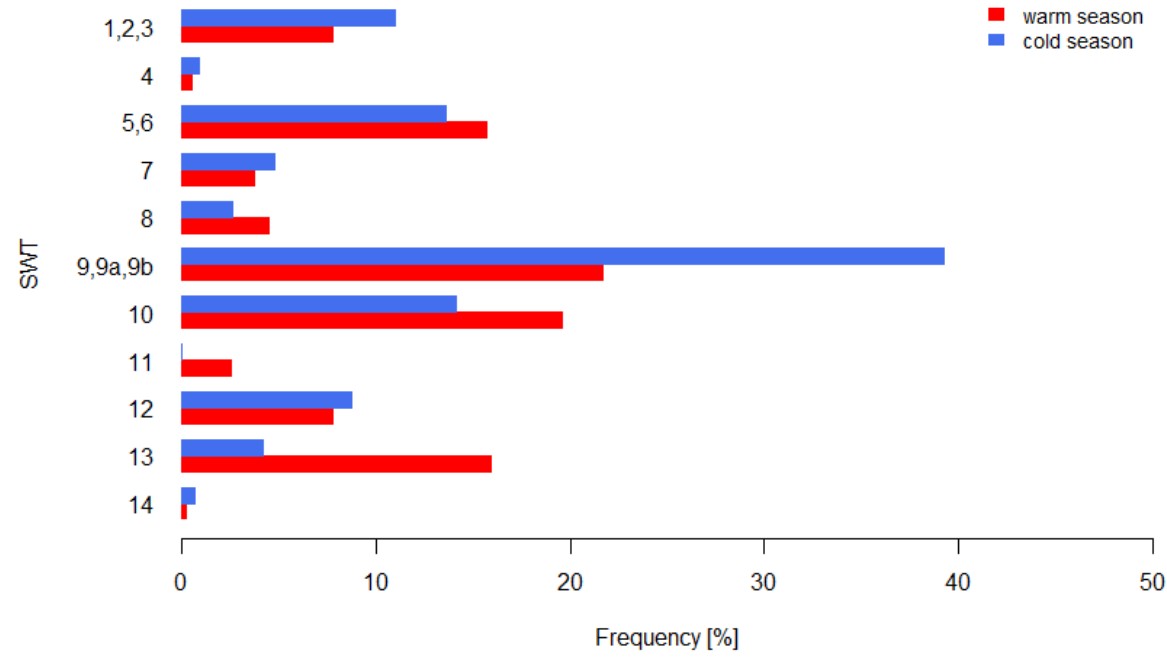

**Figure 8.** Frequency distributions of daily synoptic weather types by Bugayev's classification during the cold (Sep-Feb) and warm (Mar-Aug) seasons in the period of 1935-2014 y. Definitions of SWT can be seen in Table 1 in appendices section.





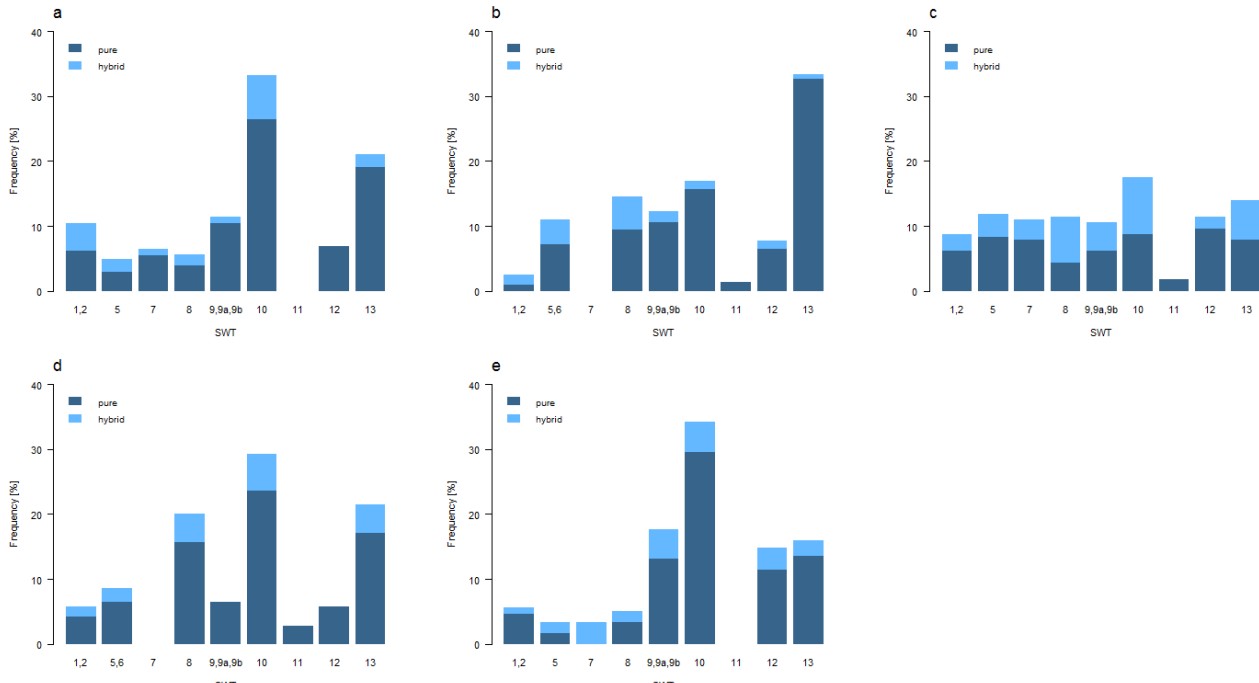

**Figure 9.** Frequency of mudflows under the synoptic weather types (SWT) over Uzbekistan during 1984-2013 (March-August) Zerafshan Basin (101 days); b) Fergana Valley (147 days); Chirchik-Akhangaran Rivers Basin (57 days); d) Kashkadarya Basin (35 days); e) Surkhandarya Basin (44 days). Definitions of SWT can be seen in Table 1 in appendices section.



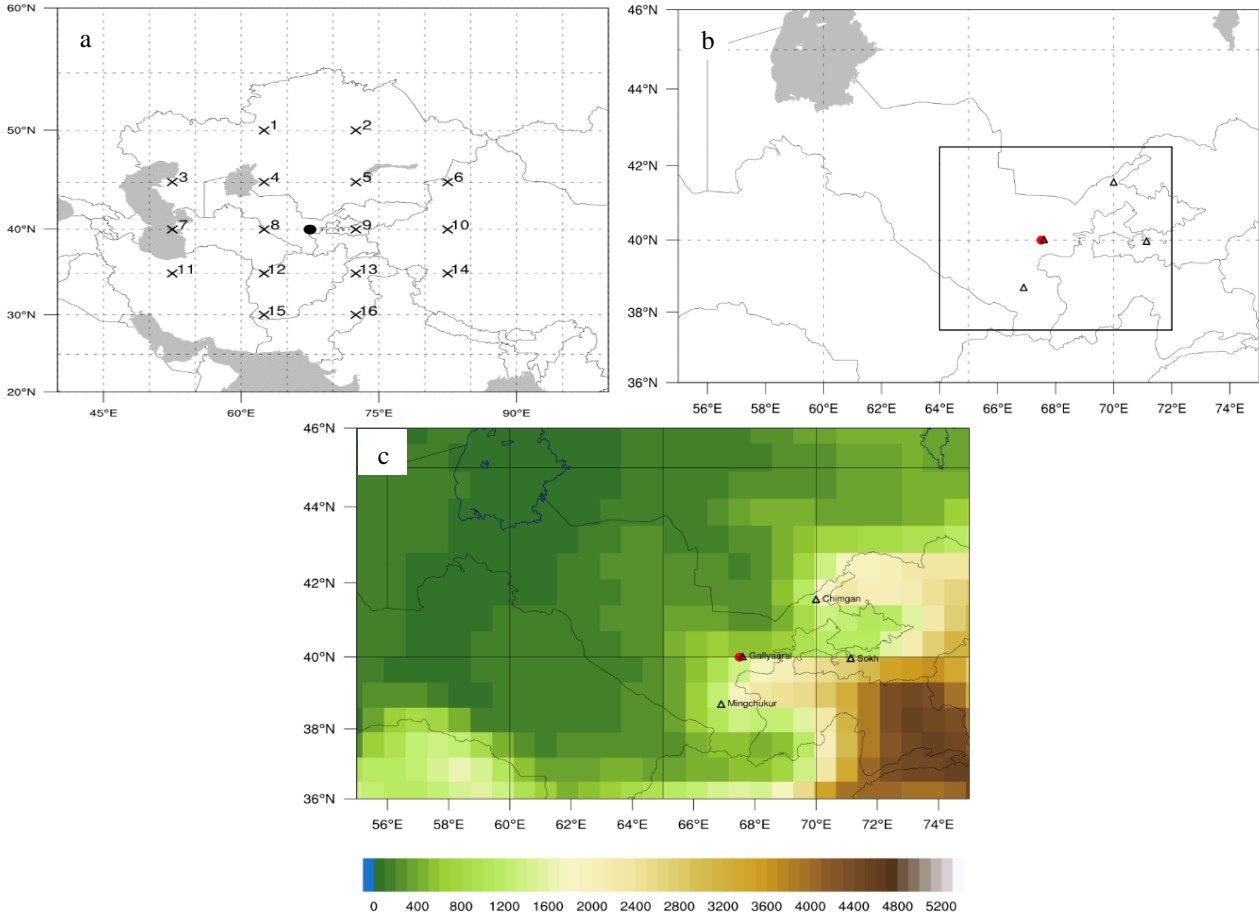

**Figure 10. a)** Location of the study domain together with the 16 grid points and central grid point (40N-67.5E) used in the automated
weather circulation type; **b)** Location of selected stations in the study area around the central grid point 40.0 N-67.5 E (circle) of
CWT objective method. Stations (triangles): Gallyaral (40.02 N-67.60 E), Chimgan (41.57 N-70.00 E), Sokh (39.97 N-71.13 E) and
Mingchukur (38.70N – 66.90 E); **c)** ERA-Interim orography map and the location of central grid point (red circle) together with
representative five stations (black triangles).





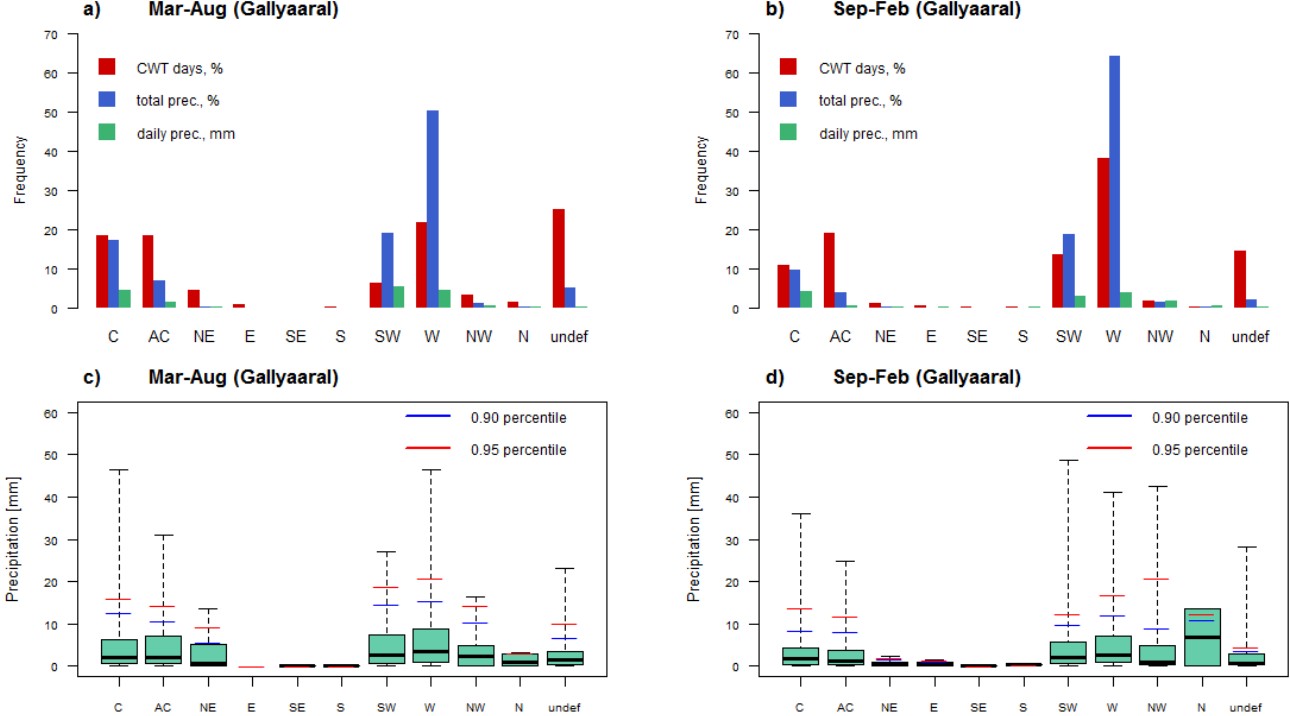

**Figure 11.** Contribution of CWT classes to the observed precipitation over Gallyaral station (Zerafshan Basin), for warm (a) and cold (b) seasons during the period 1984-2013. CWT days - frequency of each class in percentage; % total precipitation - contribution of each class to the overall precipitation; mm/day - daily average precipitation per CWT. Box plots (c, d) show the statistical analysis of daily precipitation per CWT class. The black lines in boxes represent medians for each weather type; the lower (upper) box limits mean the first (third) quartiles; the lower (upper) whiskers show the minimum and maximum values of the precipitation; blue and red lines represent 0.90[th] and 0.95[th] percentiles of the precipitation. The figure has different scales.




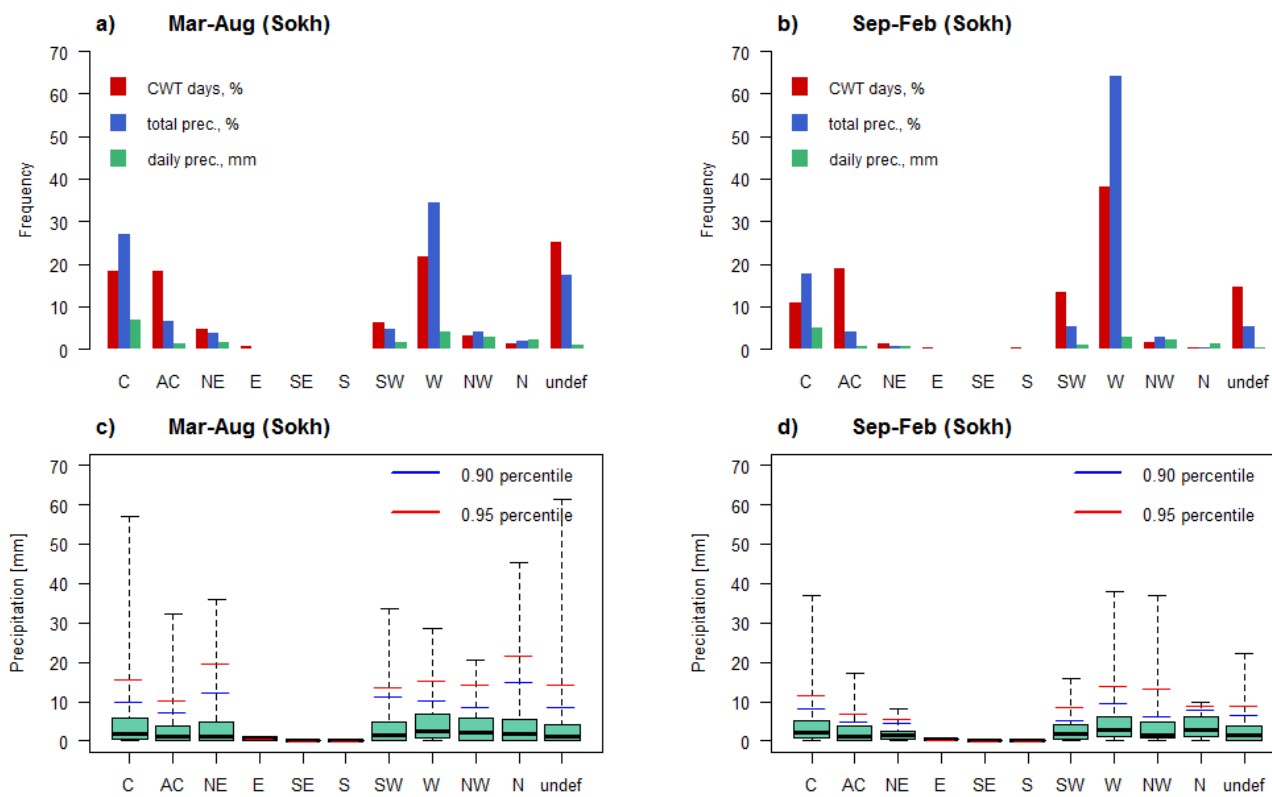

**Figure 12.** As Figure 11, but for Sokh station (Fergana Valley).



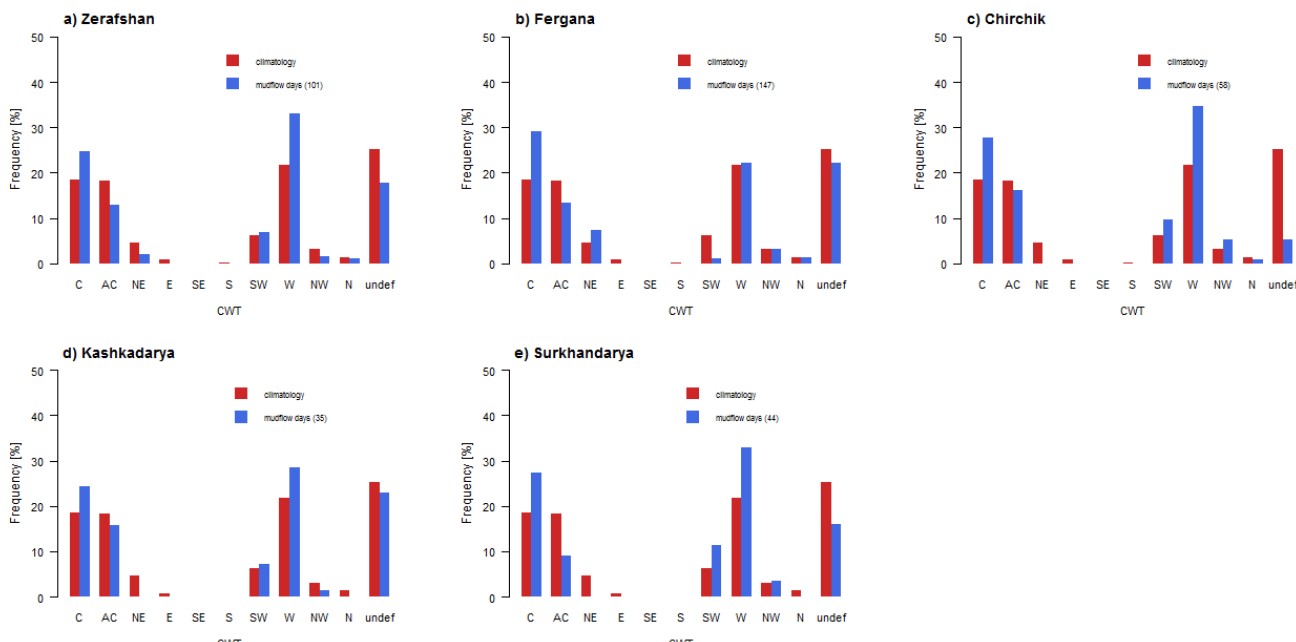

**Figure 13.** Frequency of CWT (700GPH) climatology for the period March-August, 1984-2013 (red bars) and mudflow days (blue bars) occurred in Zerafshan Basin (a), Fergana Valley (b), Chirchik-Akhangaran (c), Kashkadarya (d) and Surkhandarya (e) basins in Uzbekistan. Central grid point is 47.0 N - 67.5 E.





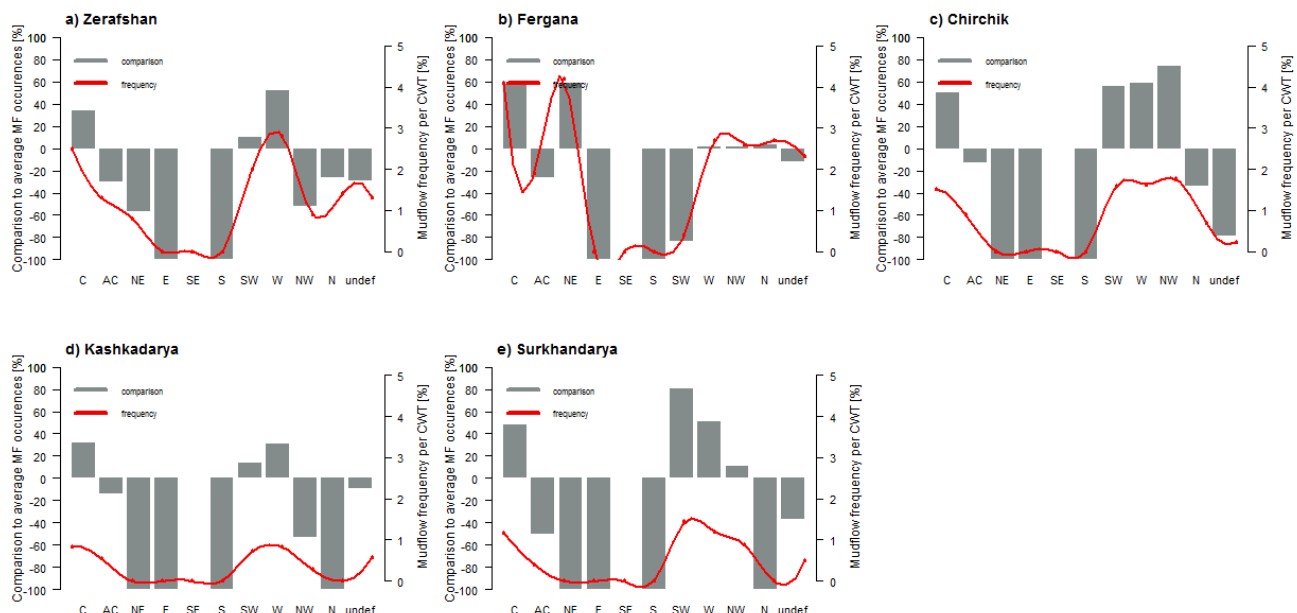

**Figure 14.** Frequency of mudflow days per class in comparison to per CWT climatology for the March-August period between 1984 and 2013 (red line, %) in five regions (Zerafshan, Fergana, Chirchik-Akhangaran, Kashkadarya and Surkhandarya) and in contrast to the average number of mudflow days per CWT (grey bars, %). For example, of the total 5520 CWT days in warm season throughout 30 years (Mar-Aug, 1984-2013), there were up to 1018 days of cyclonic circulation with the frequency of 2.5 % (25 days) which associated with mudflow events in Zerafshan Valley. Furthermore, it was identified that mudflow events occurred on cyclonic days increased up to 34% in comparison to the average value of mudflow days (1.8% or 101 mudflow days against 5520 CWT days) there.



**Figure 15.** Antecedent Daily Rainfall Model applied to the representative stations Gallyaaral (a), Chimgan (b), Mingchukur (c) and Sokh (d) for the period 1984-2013.





**Figure 16.** Threshold probabilities initiating mudflow occurrences per CWT class in Gallyaaral Station for the period of March-April 1984-2013. Black dot is day without mudflow, green triangle – days with probable mudflow, red triangle is a day initiated mudflow occurrences in the study area.







**Figure 17.** As Figure 16, but for Sokh station (Fergana Valley).



**Tables**

**Table 1.** Mudflow disasters causing fatalities and other relative damages over the period 2005-2014 in Uzbekistan (Data source: Uzhydromet)

| year | death | household or property damages | livestock counts | infrastructure damage | | | | | agricultural crops (ha) | | | |
|---|---|---|---|---|---|---|---|---|---|---|---|---|
| | | | | highway (km) | bridges | hydrologic bridges or | schools | other | cotton fields | wheat fields | gardens | other |
| 2005 | | 860 | | | 1 | | | 2 | 200 | 69 | | |
| 2006 | 7 | 175 | | | | | | 2 | 152 | 165 | 118 | 22 |
| 2007 | | 8 | 1 | 6 | 15 | 7 | | 3 | | 2 | | 6 |
| 2008 | 7 | 413 | 1 | 0.3 | 5 | | | 49 | 747 | 261 | | 123 |
| 2009 | 8 | 498 | 80 | | 14 | 5 | 2 | | 966 | 834 | 56 | 18 |
| 2010 | 8 | 41 | | | 6 | | 2 | 7 | | 5 | | 3 |
| 2011 | 2 | 94 | 50 | 0.5 | | 1 | | 52 | 483.5 | 318.6 | 0.12 | 10.1 |
| 2012 | 5 | 773 | 3 | 2.7 | 25 | 6 | 1 | 55 | | | | |
| 2013 | 1 | 31 | | 0.012 | 2 | 6 | | 3 | | | | 200 |
| 2014 | | | | | | | | 4 | | | | |
| total | 38 | 2893 | 135 | 10 | 68 | 25 | 5 | 177 | 2548 | 1655 | 174 | 382 |





**Table 2.** Monthly distribution of mudflow events in five regions in Uzbekistan.

| Basin | Data | Jan | Feb | Mar | Apr | May | Jun | Jul | Aug | Sep | Oct | Nov | Dec | Total | Mean |
|---|---|---|---|---|---|---|---|---|---|---|---|---|---|---|---|
| Fergana | 1875-2014 | 8 | 21 | 34 | 292 | 605 | 462 | 218 | 67 | 2 | 0 | 2 | 0 | 1711 | 12 |
| Zerafshan | 1872-2014 | 7 | 36 | 80 | 177 | 157 | 54 | 19 | 10 | 0 | 2 | 1 | 1 | 544 | 3.8 |
| Surkhandarya | 1890-2014 | 0 | 0 | 20 | 151 | 146 | 63 | 7 | 0 | 0 | 2 | 0 | 0 | 389 | 3 |
| Chirchik-Akhangaran | 1870-2014 | 1 | 3 | 31 | 91 | 64 | 17 | 19 | 7 | 0 | 0 | 0 | 0 | 233 | 1.6 |
| Kashkadarya | 1877-2014 | 0 | 3 | 12 | 92 | 42 | 9 | 0 | 6 | 0 | 1 | 0 | 0 | 165 | 1 |
| Total | 1870-2014 | 16 | 63 | 177 | 803 | 1014 | 605 | 263 | 90 | 2 | 5 | 3 | 1 | 3042 | 21 |





**Table 3.** Threshold probabilities (10%, 50% and 90%) inducing mudflow events in 4 stations (Gallyaaral, Chimgan, Mingchukur and Sokh); $r_a$ - antecedent rainfall index (mm), r - daily rainfall value (mm).

| Station | 10% | | 50% | | 90% | |
|---|---|---|---|---|---|---|
| | $r_a$ | $r$ | $r_a$ | $r$ | $r_a$ | $r$ |
| Gallyaaral | $\leq 40$ | $\leq 20$ | $\leq 90$ | $\leq 40$ | $\leq 130$ | $\leq 60$ |
| Chimgan | $\leq 90$ | $\leq 45$ | $\leq 110$ | $\leq 75$ | $\leq 125$ | $\leq 100$ |
| Mingchukur | $\leq 60$ | $\leq 25$ | $\leq 85$ | $\leq 50$ | $\leq 100$ | $\leq 65$ |
| Sokh | $\leq 30$ | $\leq 10$ | $\leq 70$ | $\leq 35$ | $\leq 115$ | $\leq 50$ |

**Table 4.** Rainfall threshold probability equations of mudflow occurrences in selected areas (P – probability, r – daily rainfall, $r_a$- antecedent rainfall, Pr(>Chi) - Chi-squared results to the regions respectively).

| Station | Probability equation | Pr(>Chi) |
|---|---|---|
| Gallyaral | $\log\left(\frac{P}{1-P}\right) = -3.87 + 0.10 * r + 0.05 * r_a$ | 9.402e-05 |
| Chimgan | $\log\left(\frac{P}{1-P}\right) = -5.84 + 0.082 * r + 0.099 * r_a - 0.0015 * r_a^2 + 0.00001 * r_a^3$ | 1.455e-10 |
| Mingchukur | $\log\left(\frac{P}{1-P}\right) = -4.54 + 0.89 * r - 0.003 * r_a + 0.0008 * r_a^2$ | 8.036e-07 |
| Sokh | $\log\left(\frac{P}{1-P}\right) = -3.50 + 0.11 * r + 0.05 * r_a$ | 3.285e-07 |



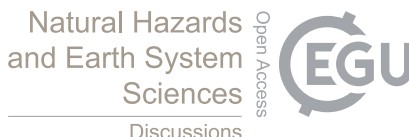
**Appendices**

**Table A1.** Synoptic weather types (SWT) of Central Asia and general weather characteristics over the region and in Uzbekistan. SWT 1-9, 9a and 10-11 was classified by Bugayev et al. (1957); types 9b and 12-15 were added later by the researchers of the Hydrometeorological Scientific Institute in Uzbekistan.

| Group | Code | Synoptic Weather Type (SWT) | Source region | Air mass | Characteristics |
|---|---|---|---|---|---|
| **Group A** Cyclones from south and south-west | 1 | South Caspian Cyclone | Southern part of Caspian Sea, east of Mediterranean Sea, Mesopotamia, Northern part of Arabian Peninsula | tropical | rising temperature in the warm sector of the cyclone in winter, heavy precipitation especially in mountain areas, strong winds, dust storms along the desert areas |
| | 2 | Murgab Cyclone | Cyclone forms as a wave in Iraq, Mesopotamia, Iran and approaches to the Southern part of Turkmenistan, Murghab and Tejen Rivers' basins | tropical, continental (temperate zones) | mild and wet in the warm sector of the cyclone, strong winds, heavy precipitation, thunderstorms in spring, floods in the rivers, sometimes radiation fog |
| | 3 | Upper-Amudarya Cyclone | Afghanistan, West Pakistan, Persian Gulf | tropical | warm air flow, cloudy; precipitation may be observed in the territory of Tajikistan; strong winds in the mountain areas, sometimes fog |
| | 4 | Broad carrying the warm air | South-westerly and southerly flows in the troposphere approach to the Southern part of European Russia, Western Kazakhstan and Central Asia | tropical, continental (temperate zones) | warm, clear, dry weather condition with light winds |
| **Group B** Advection of cold airflow from north and north-west | 5 | Advection of cold north-westerly air flow | South-eastern part of European Russia, Western Kazakhstan and Ustyurt | arctic (Siberian sector), continental (temperate zones) | mainly cold in winter with cloudiness, precipitation and strong winds; in summer cool weather, precipitation depends on the orography and convection process; frosts in spring and autumn |
| | 6 | Advection of cold northerly air flow | Ural, Western Siberia, Kazakhstan | arctic (Greenland and the North Sea), continental (temperate zones) | very cold, sometimes severe weather with winds, little precipitation and fog in winter season; thundery and rainfall in summer in the mountain areas; frost in early spring and late autumn |
| | 7 | Synoptic wave activity on a cold front | Eastern Mediterranean, Middle East | Mediterranean, Atlantic (subtropical) | mostly the weather is wet with changeable temperature, sleet showers, winds, occasionally thundery in spring |
| | 8 | Stationary Cyclone over the Central Asia | Regeneration of western or south-western cyclones | Mediterranean, Atlantic, tropical | sleet in cold period; in summer cool and heavy rainfall especially in south-eastern mountain areas, thundery, slight winds |
| | 10 | Advection of westerly air flow | Central and Southern Europe (westerly moist) European Russia and Ukraine (westerly cold) | Atlantic and continental (temperate zones), sometimes arctic | cool, strong winds, precipitation in cold period; in spring and early summer the weather is wet giving most rain, temperature falling, dust storms, thundery |
| | 15 | Diving Cyclone | Norwegian Sea, Barents Sea, Kara Sea | arctic | precipitation, strong winds |
| **Group C** Anticyclonic weather | 9 | South-western periphery of Anticyclone | Siberian High | arctic | in general, it is clear and mostly dry with slight winds; radiation fog in the piedmont and mountain areas in the first phase of the synoptic type, cloudiness and precipitation might be observed in eastern mountain areas of Central Asia |
| | 9a | South-eastern periphery of Anticyclone | Stationary anticyclone over the Ustyurt, lower Volga or western Kazakhstan (part of Siberian High) | arctic | clear, cold, frost and mist on the plain surface, sometimes precipitation in mountain areas in cold period; in summer, it is cool and slight winds |
| | 9b | Southern periphery of Anticyclone | Siberian High extends to the eastern regions 50-55° N in which Central Asia is on its southern periphery | arctic | mostly this synoptic weather type is cold and dry with foggy days in winter phase; in summer the weather is cool and clear |
| | 11 | Summer thermal low | Non frontal low pressure area over southwest Asia | tropical dry | clear, dry, very hot, haze, winds, dust storms |
| | 12 | High level of small barometric gradient | The area of high pressure over the Central Asia which units the Siberian High and the anticyclone over the European Russia | mostly arctic | generally cold, dry and clear weather with light winds; in cold periods it is foggy and little precipitation in south-eastern regions |
| | 13 | Low level of small barometric gradient | Low-pressure area over the Central Asia which aligns in meridional orientation | moderate zones | in most parts of Central Asia there are dry and warm weather conditions; in winter the weather is mostly foggy with slight precipitation; in summer convective clouds and heavy rainfall may be observed in local areas |
| **Group D** Mid-latitude cyclone | 14 | Western Cyclone | Cyclone tracks from the Mediterranean Sea, sometimes from Northern Africa to the Black Sea or Middle East then reaches to the Central Asia through the Caspian Sea | Mediterranean | strong winds, dust storm in the desert area, precipitation especially in mountain areas |

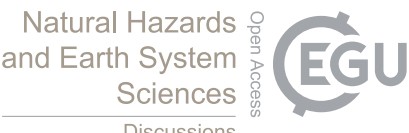

**Table A2.** Threshold probability (10%, 50% and 90%) values initiating mudflow episodes per CWT for individual stations in Uzbekistan; $r_a$ - antecedent rainfall index (mm), r - daily rainfall value (mm).

| CWT | Gallyaaral | | | | | | Chimgan | | | | | | Mingchukur | | | | | | Sokh | | | | | |
|---|---|---|---|---|---|---|---|---|---|---|---|---|---|---|---|---|---|---|---|---|---|---|---|---|
| | 10% | | 50% | | 90% | | 10% | | 50% | | 90% | | 10% | | 50% | | 90% | | 10% | | 50% | | 90% | |
| | $r_a$ | $r$ | $r_a$ | $r$ | $r_a$ | $r$ | $r_a$ | $r$ | $r_a$ | $r$ | $r_a$ | $r$ | $r_a$ | $r$ | $r_a$ | $r$ | $r_a$ | $r$ | $r_a$ | $r$ | $r_a$ | $r$ | $r_a$ | $r$ |
| NE | - | - | - | - | - | - | - | - | - | - | - | - | - | - | - | - | - | - | ≤10 | ≤0.1 | ≤70 | ≤20 | ≤120 | ≤45 |
| SW | ≤40 | ≤20 | ≤70 | ≤45 | ≤100 | ≤60 | ≤80 | ≤80 | ≤85 | ≤100 | ≤90 | ≤115 | ≤70 | ≤30 | ≤85 | ≤50 | ≤95 | ≤70 | ≤40 | ≤18 | ≤65 | ≤30 | ≤95 | ≤42 |
| W | ≤40 | ≤20 | ≤70 | ≤40 | ≤95 | ≤55 | ≤95 | ≤50 | ≤115 | ≤70 | ≤130 | ≤100 | ≤60 | ≤30 | ≤85 | ≤58 | ≤108 | ≤78 | ≤25 | ≤18 | ≤50 | ≤40 | ≤75 | ≤55 |
| NW | - | - | - | - | - | - | ≤60 | ≤20 | ≤70 | ≤40 | ≤75 | ≤65 | - | - | - | - | - | - | ≤40 | ≤10 | ≤60 | ≤20 | ≤85 | ≤30 |
| N | - | - | - | - | - | - | - | - | - | - | - | - | - | - | - | - | - | - | ≤60 | ≤10 | ≥120 | ≤35 | ≥120 | ≤55 |
| C | ≤30 | ≤20 | ≤60 | ≤50 | ≤90 | ≤70 | ≤130 | ≤40 | ≥140 | ≤75 | ≥140 | ≤105 | ≤58 | ≤30 | ≤78 | ≤60 | ≤90 | ≤88 | ≤25 | ≤10 | ≤60 | ≤30 | ≤100 | ≤50 |
| AC | ≤40 | ≤15 | ≤105 | ≤40 | ≤140 | ≤70 | ≤85 | ≤40 | ≤100 | ≤60 | ≤110 | ≤80 | ≤78 | ≤25 | ≤105 | ≤40 | ≤130 | ≤55 | ≤40 | ≤10 | ≤85 | ≤28 | ≤120 | ≤40 |
| und | - | - | - | - | - | - | ≤90 | ≤120 | - | - | - | - | ≤60 | ≤15 | ≤90 | ≤25 | ≤120 | ≤40 | ≤55 | ≤8 | ≥120 | ≤35 | ≥120 | ≤58 |





**Figure A1.** CWT south-westerly (a), westerly (b), north-westerly (c), northerly (d), cyclonic (e), anticyclonic (f) and undefined 5 (g) weather types characteristics on the mudflow days occurred in Chirchik-Akhangaran Basin for the period 1984-2013. ERA-Interim 700 hPa geopotential height, relative humidity and wind component were used to produce this figure.



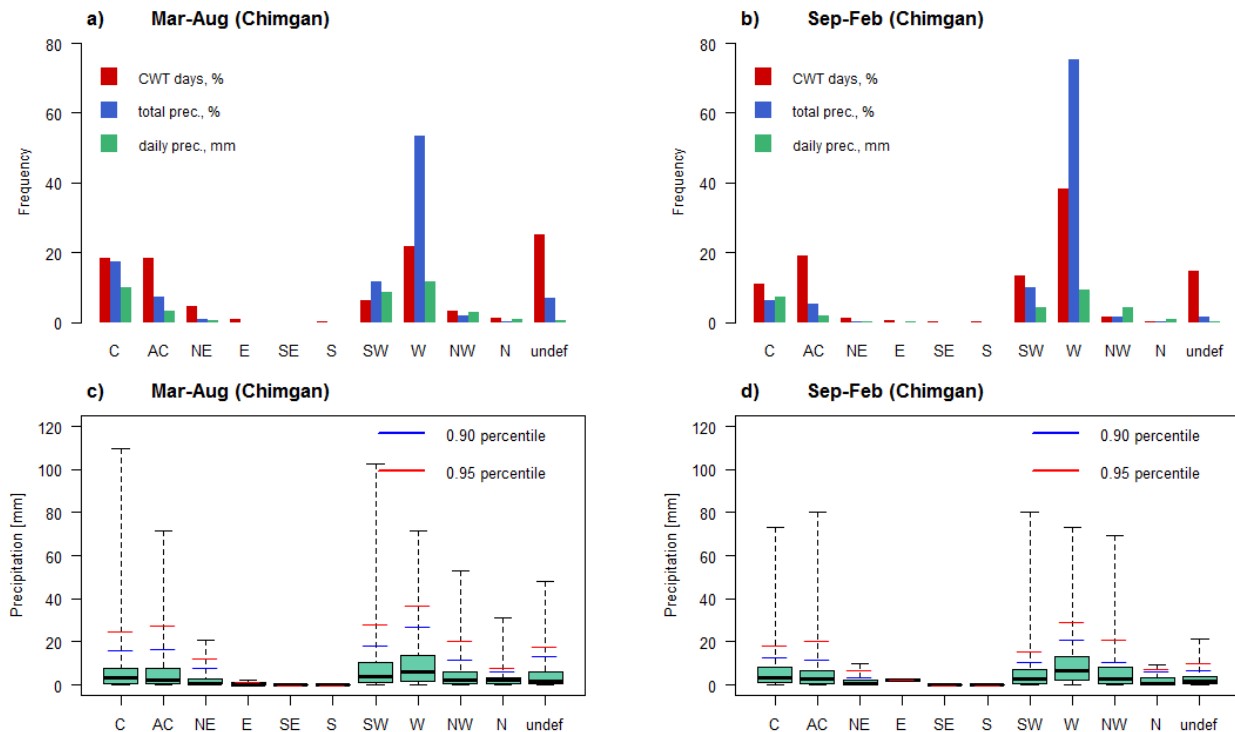

**Figure A2.** As Figure 11, but for Chimgan station (Chirchik-Akhangaran Basin)





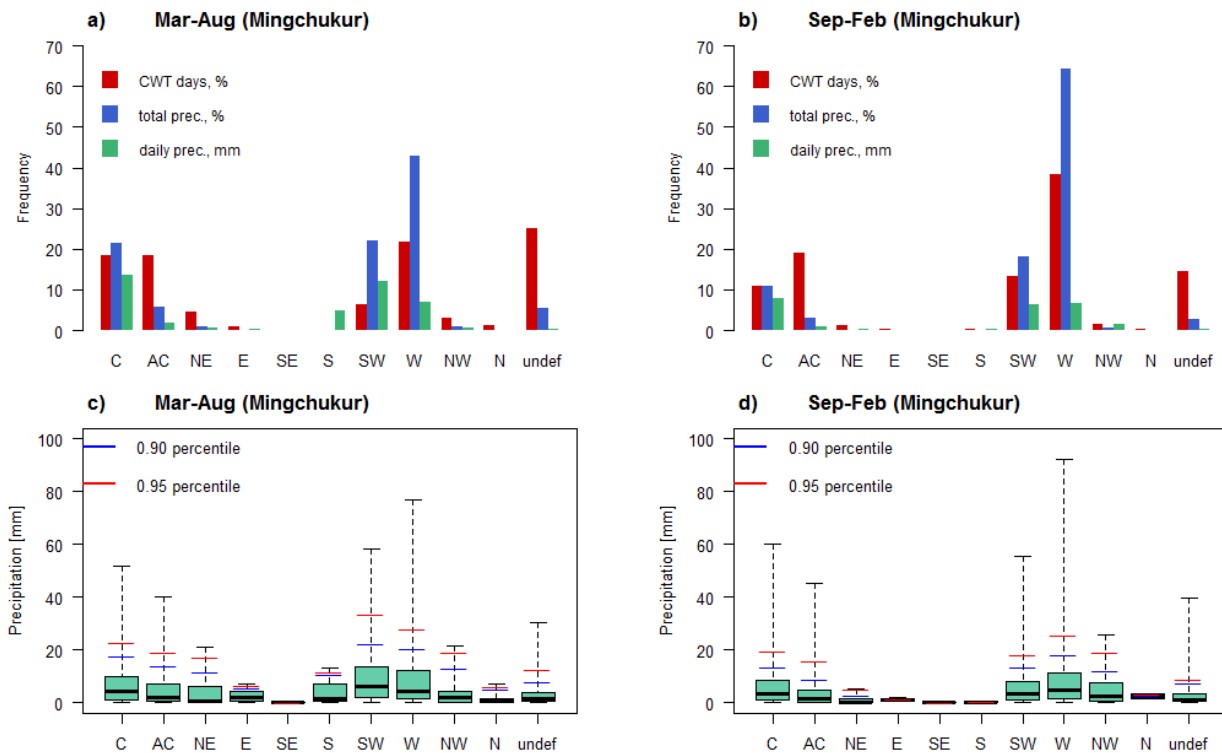

**Figure A3.** As Figure 11, but for Mingchukur station (Kashkadarya Basin)





**Figure A4.** As Figure 16, but for Chimgan station (Chirchik-Akhangaran Basin)



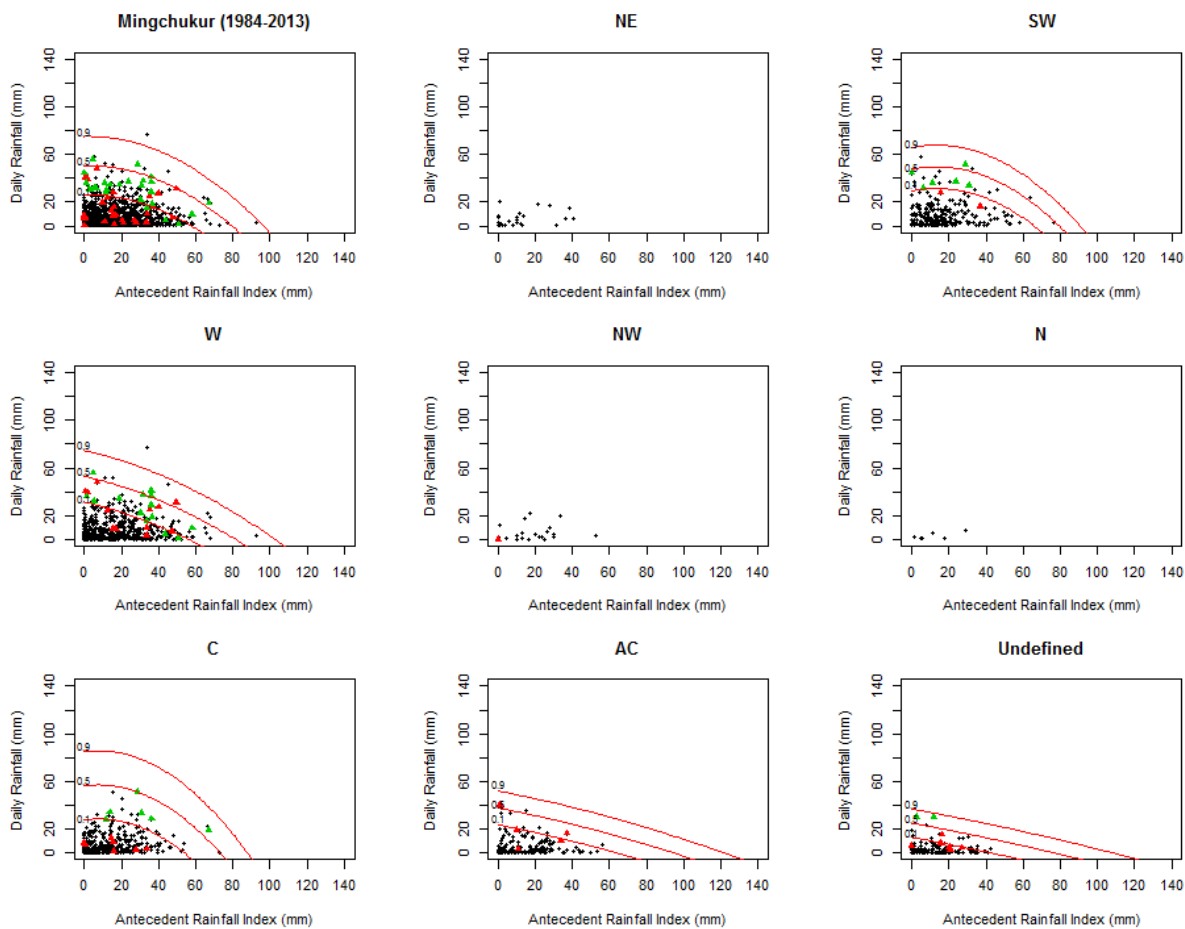

**Figure A5.** As Figure 16, but for Mingchukur station (Kashkadarya Basin)