# Peer review of "The Role of Synoptic Processes for Mudflow Formation in the Piedmont Areas of Uzbekistan"

_Natural Hazards and Earth System Sciences, 2018_

## Referee Comment (RC1) · Anonymous Referee #1 · 25 May 2018

This is a study of the relationship between atmospheric factors, based on the classification of weather types, and the mudflows. The study is performed with the historical registers of mudflows, and the pressure data from ERA-Interim reanalysis for the classification of weather types and the air flow patterns already known. The relationship between the two variables is studied using the frequency of the mudflows for each of the weather types. In addition, a statistical model (Antecedent Daily Rainfall Model) to evaluate the probability of daily rainfall resulting into mudflow.

This study meets the required standards of the journal, since it addresses relevant

questions, it uses a high-quality database, it applies adequate statistical methods, provides a correct interpretation of the results properly supported with bibliographic references and it has a correct structure of contents and writing style.

However, the number of figures and tables seems to be unnecessarily large, with images of low quality.

The are some minor details that need to be corrected: - Authors must indicate the meaning of the acronyms used in the manuscript (e.g., page 3 AOGCM and RCM) - A uniform way to reference figures must be used (Fig or Figure) - Figure 1 has very low quality. It would also be advisable to provide coordinates or/and additional localization map. This map is difficult to interpret. Authors should explain that it represents the political-administrative division of the country together with the 5 locations of study. - Figure 3 should indicate the average monthly temperature. - Authors should consider the possibility of reducing the number of figures and tables (e.g., figure 4 and table 1 can be removed). - Figure 7 should indicate for latitude/longitude the symbol of degrees and North and East. - In the caption for Figure 9 the reference to subfigures a and c is missing. - It is difficult to interpret graphs 8 and 9 with numbers of SWT, and then its link to table A1 (annex). I advise the authors to use abbreviations that allow to easily remember the synoptic weather types instead of numbers. - Figure 10 has errors in it. The grey backgrounds in 10a and 10b don't indicate anything, so they should be removed. Moreover, it's advisable to show the area of study with a grey background. In fig. 10c the measurement unit (m) should be indicated for the orography. - Authors should include in the text references to figures 10b, A1 and to table A2. - Figure 14 has a misleading description. It is not entirely clear what graphs represent.

---

## Referee Comment (RC2) · Anonymous Referee #2 · 30 Jun 2018

After the careful reading of the manuscript "Statistical Characteristics of Mudflows in the Piedmont Areas of Uzbekistan and the Role of Synoptic Processes for their Formation" I believe that this is a good contribution to the study of the synoptic processes and its relation to the mudflow occurrences in Uzbekistan. Also, a statistical model (Antecedent Daily Rainfall Model) to evaluate the probability of daily rainfall resulting into mudflow was also produced. This study represents a good contribution to the understanding of the SWT and CWT associated with the mudflows events and the ADRM to evaluate the probability of a daily rainfall to produce a mudflow in the study area.

[Figure]

This study is definitely in the scope of NHESS Despite of these studies are not completely innovative, this work is interesting and its methodology can be applied for other study areas than Uzbekistan. I recommend moderate revisions and my advices and reasons are listed below. 1. Scientific data and results sometimes are not presented in a very clear way, and the manuscript structures sometime mixes state of art text with results, which make the ideas difficult to understand. 2. Presented results are sufficient to support conclusions but I would recommend improve the conclusions section with some discussion about the data sources issues (e.g. debris flow database, daily precipitation, CWT) that the authors faced in this study and that can be important to the use of this methodology in other study areas. 3. Number and quality of figures/tables – I strongly recommend to reduce the number of figures. 4. English language deserves a moderate revision. 5. The title includes statistical characteristics of mudflows but in practice the main purpose of this work is to relate mudflows with the role of synoptic processes. I suggest removing the statistical characteristics from the title. 6. In the introduction section it misses geological characteristics of the country in order to understand the mud flow distribution, conditioning and triggering factors. Also the last paragraph of the introduction section is not mandatory to understand the work, so I suggest removing this paragraph. 7. Section 2.1 – Data – in this section is not clear how many meteorological stations will be used in this study and which one is located in the mountains and which one is located in foothill zones. This must be clear as also the data period of each station. 8. Section 2.2 – Methods – in this section misses a subtopic concerning the SWT. This text can be added from page 8, section 4.1, where authors mix the state of art of SWT with the results. With this change the methods text will include all methods used in this work. In this section it must be explained how the mudflows around the climatological stations were selected. It was used some kind of buffer or other criteria? Please explain this in the text. 9. Page 5, first paragraph – When authors refer some examples of works that used the calibrated antecedent rainfall model, the reference Zêzere et al (2015) did not used this method as you can verify in the original source. It was used in a previous work by "Zêzere and Rodrigues (2002)

Rainfall thresholds for landsliding in Lisbon Area (Portugal). In: Rybar J, Stemberk J, Wagner P(eds) Landslides. Lisse, A. A. Balkema, pp 333–338". 10. Section 3.1 – general climate conditions – second paragraph – "the long term climatological shows. . .." – please refer how many years correspond to this long period of climatological data. 11. Page 6, line 25 – "The spatial distribution of the average precipitation. . ." – again please refer the period of this data. 12. Section 3.2 – mudflows in Uzbekistan – you should specify which are the data sources of the archive data of mudflows. 13. Page 7 line 26 – why do you use the terms episodes and events of mudflows? It has the same meaning? Please clarify this topic in the text. 14. Authors relate the highest peaks of mudflows in the 1930s, 1960s and 1990s. I ask if these peaks can be related with some land use changes occurred in the country. 15. Sections 4 and 5 are more related to the results presentation, so the text needs to be improved and better structured. In the section 4.1 the text must be limited to the results and the part related to the methods can be included in section 2.

Notes about the figures: This work should reduce the number of figures and tables. 1. Figure 1 – it misses a scale and a north arrow in the map, and also the name of the neighboring countries. 2. Figure 2 – figure caption is confusing. It can be only methodological flowchart. 3. Figure 3 - these stations are located in which basin? 4. Figure 4 – this map has several cartographical problems. The legend and the north arrow should be reduced because its size is over exaggerated in comparison to the map size. Erase the title "distribution of precipitation" because the figure caption will detail that information. Again it misses the scale in this map. This figure caption can be simply "total annual precipitation. . ..." because there is no need to repeat that this is a map. 5. Figure 6 – the figure caption can be only "Monthly mudflow frequencies. . ." 6. Figure 7 – it misses the legend of the codes used in this figure in order to be understandable. 7. Figure 8 – please specify for which area corresponds this graph. 8. Figure 10 – b) it misses the legend of the triangles or put the name of the stations in the figure. C) include the units of the legend (Fig10c)and remove the blue from the legend because it has no cartographical representation. 9. Figure 11, 12 and A2 and A3 can

be joined in two figures. In the first figure you can put a) and b) for the 4 study areas (one page) and in the second one you can put c) and d) for the 4 study areas (one page). With this solution you reduce the number of figures and also the reference of these figures in the text will be clearer. Also reduce the figure caption of these figures, removing some text that can be referred in the text. 10. Figure 14 – reduce the length of the figure caption because it is too long and include the example in the text, not in the caption. 11. Figure 15 – include the meaning of the red lines in the graphs. 12. Figure 16, 17, A5 and A6 in my opinion are not necessary to understand the results. Please consider remove these figures or other alternative to reduce the number of figures.

Notes about the tables: 1. Table 1 – it is necessary to put the unit of each row of the table.

---

## Author Comment (AC1) · 13 Jul 2018

**Answer to the Anonymous Referee #1 comments**

Dear Referee

We highly appreciate your comments and suggestions adding valuable inputs to improve our manuscript. Based on your insightful guidance, we revised our paper and made respective minor changes. We hope our revision meet your approval.

**Comment:** Authors must indicate the meaning of the acronyms used in the manuscript (e.g., page 3 AOGCM and RCM)

**Response:** In the revision, we indicate these acronyms as Atmosphere-Ocean Global Circulation Models (AOGCM) and Regional Climate Models (RCM) on page 3 of the paper. We checked all other acronyms and hope all are explained now.

**Comment:** A uniform way to reference figures must be used (Fig or Figure)

**Response:** We have gone through the paper and included reference to figures as (Figure)

**Comment:** Figure 1 has very low quality. It would also be advisable to provide coordinates or/and additional localization map. This map is difficult to interpret. Authors should explain that it represents the political-administrative division of the country together with the 5 locations of study

**Response:** We produced a new, high quality figure showing the main mudflow regions and administrative divisions respectively. We included mudflow locations for the 2005-2014 period to highlight the different regions.

[Figure]

**Figure 1.** Mudflow occurrences for the years 2005-2014 in areas with high probability of mudflow passage in Uzbekistan: Zerafshan Basin (blue dots) in central part of the country; Fergana Valley (red dots) in the east; Chirchik-Akhangaran Basin (orange) in the north-east; Surkhandarya (green) and Kashkadarya (violet) rivers' basins in the south of Uzbekistan. Map also represents political administrative divisions and administrative centers/cities of the country. Map does not include inland water resources.

**Comment:** Figure 3 should indicate the average monthly temperature.

**Response:** Figure 3 actually indicates the average monthly temperature (black curve line) and precipitation (grey bars) in original version, however we have changed the colour to increase visibility.

[Figure]

**Figure 3.** The 30-year means (1984-2013) of monthly temperature (°C, red line) and precipitation (mm, green bars) in four selected stations namely Gallyaral in Zerafshan Basin (a), Chimgan in Chirchik-Akhangaran Basin (b), Mingchukur representing Kashkadarya and Surkhandarya Basins (c) and Sokh in Fergana Valley (d) with high occurrences of mudflow in Uzbekistan. Graphs have different scales

**Comment:** Authors should consider the possibility of reducing the number of figures and tables (e.g., figure 4 and table 1 can be removed).

**Response:** We followed this recommendation and removed Figure 4 and moved Table 1 into the Appendix.

**Comment:** Figure 7 should indicate for latitude/longitude the symbol of degrees and North and East.

**Response:** Figure 7 was presented as originally published. However, we produced a figure based on the reference material and added information about the nature of the flow.

[Figure]

**Figure 7.** Scheme of synoptic weather types in Central Asia and Uzbekistan during the cold (a) and warm seasons (b) of the year (after Inagamova et al., 2002). Blue and red cursors indicate relatively cold and warm air trajectories approaching to the investigation area (grey background). Abbreviations and numbers of each weather type explained in Table 1 of Appendices.

**Comment:** In the caption for Figure 9 the reference to subfigures a and c is missing.

**Response:** Many thanks. Corrected.

**Comment:** It is difficult to interpret graphs 8 and 9 with numbers of SWT, and then its link to table A1 (annex). I advise the authors to use abbreviations that allow to easily remember the synoptic weather types instead of numbers.

**Response:** We are appreciative of this comment. However, in the original work to SWT each weather type is given as a number code. We would like to conserve this. Nevertheless, we added abbreviations in order to indicate the respective weather class in the revised version.

[revised manuscript text omitted]

**Comment:** Figure 10 has errors in it. The grey backgrounds in 10a and 10b don't indicate anything, so they should be removed. Moreover, it's advisable to show the area of study with a grey background. In fig. 10c the measurement unit (m) should be indicated for the orography

**Response:** Many thanks for this point. Grey background in Figures 10a and 10b was changed to a blue colour to indicate inland water resources. Investigation area was filled in grey background. Orography unit (m) in Figure 10c was added.

[Figure]

**Figure 10. a)** Location of the study domain together with the 16 grid points and central grid point (40N-67.5E, red circle) used in the automated weather circulation type; **b)** Investigation area shown in rectangle and location of selected stations (black triangles) around the central grid point 40.0 N-67.5 E (red circle) of CWT objective method. Stations: Gallyaral (40.02 N-67.60 E), Chimgan (41.57 N-70.00 E), Sokh (39.97 N-71.13 E) and Mingchukur (38.70N – 66.90 E); **c)** ERA-Interim orography map and the location of central grid point (red circle) together with representative four stations (black triangles).

**Comment:** Authors should include in the text references to figures 10b, A1 and to table A2.

**Response:** Done.

**Comment:** Figure 14 has a misleading description. It is not entirely clear what graphs represent.

**Response:** Many thanks for the comment and for suggesting this improvement. We revised Figure14 adding more precisely figure caption and we also proposed to make minor correction to the figure which is detailed in the explanation.

[Figure]

**Figure 14.** Anomaly of mudflow days per CWT class (grey bars, grey axis, %) and CWT classes for mudflow days (red line, red axis, %) for the March-August period between 1984 and 2013 in five regions: a) Zerafshan Basin (101 days); b) Fergana Valley (147 days); c) Chirchik-Akhangaran Rivers Basin (57 days); d) Kashkadarya Basin (35 days); e) Surkhandarya Basin (44 days). Figure has different scales.

Explanation of the Figure 14.

Red line is CWT for mudflow days: 1018 days from total 5520 climatological CWT days in warm season associated with cyclonic circulation over Zerafshan Basin and only 25 days or 2.5 % from total cyclonic days (red line and red scale) resulted mudflow occurrences in Zerafshan (Figure 14a). Similarly mudflow anomalies or values compared to the average occurrence of mudflows (1.8% or 101 days out of 5520) in this region increased nearly 35% on cyclonic days (grey bar and grey scales). However, frequency of anticyclonic days shows 1.3 % (13 anticyclonic mudflow days out of 1012 anticyclonic totals) and decreases more than 20% compared to the average mudflow occurrences. During the investigation period E, SE and S classifications highly unlikely resulted mudflow events in Zerafshan subsequently representing missing values in figure.

---

## Author Comment (AC2) · 13 Jul 2018

**Answer to the Anonymous Referee #2 comments**

Dear Referee,

We are grateful for your comments and suggestions adding valuable inputs to improve our manuscript. Based on your pertinent remarks, we revised our paper and made respective changes and improvements. We think the revised version will address all of your points as well as the comments made by reviewer 1.

*1. Scientific data and results sometimes are not presented in a very clear way, and the manuscript structures sometime mixes state of art text with results, which make the ideas difficult to understand.*

**Response:** Following your suggestions we clarified the manuscript structure respectively.

*2. Presented results are sufficient to support conclusions but I would recommend improve the conclusions section with some discussion about the data sources issues (e.g. debris flow database, daily precipitation, CWT) that the authors faced in this study and that can be important to the use of this methodology in other study areas.*

**Response:** We agree with your remark. We would like to improve the conclusion section with discussion regarding data source issues you mentioned above.

*3. Number and quality of figures/tables – I strongly recommend to reduce the number of figures.*

**Response:** We support the referee's assertion that number of graphs should be reduced in revised version of the manuscript. We removed Figure 4 from the manuscript. We joined Figures 11 and 12 as well as A2 and A3 into two figures. We also produced one panel plot joining the figures 16-17 and A4-A5.

*4. English language deserves a moderate revision.*

**Response:** Many thanks for the comment. We will carefully check the manuscript again and improve the language in the revised version.

*5. The title includes statistical characteristics of mudflows but in practice the main purpose of this work is to relate mudflows with the role of synoptic processes. I suggest removing the statistical characteristics from the title.*

**Response:** We agree with the referee to remove the words "Statistical characteristics" from the manuscript title. The new title will be "The Role of Synoptic Processes for Mudflow Formation in the Piedmont Areas of Uzbekistan"

*6. In the introduction section it misses geological characteristics of the country in order to understand the mud flow distribution, conditioning and triggering factors. Also the last paragraph of the introduction section is not mandatory to understand the work, so I suggest removing this paragraph.*

**Response:** Geological characteristics of the basins was shortly described on page 7, section 3.2. Mudflows in Uzbekistan, lines 14-20. However, we consider your suggestion and we will add description about country's geological characteristics in the manuscript introduction. We also remove the last paragraph from the introduction section.

***7.*** *Section 2.1 – Data – in this section is not clear how many meteorological stations will be used in this study and which one is located in the mountains and which one is located in foothill zones. This must be clear as also the data period of each station.*

**Response***:* We will include additional information regarding stations data period, location and elevation in the respective section of the paper.

***8.*** *Section 2.2 – Methods – in this section misses a subtopic concerning the SWT. This text can be added from page 8, section 4.1, where authors mix the state of art of SWT with the results. With this change the methods text will include all methods used in this work. In this section it must be explained how the mudflows around the climatological stations were selected. It was used some kind of buffer or other criteria? Please explain this in the text.*

**Response***:* We will add text to the Methods section related SWT by removing text from the page 8 section 4.1. We will also explain how we selected the mudflows around four climatological stations for this manuscript.

***9.*** *Page 5, first paragraph – When authors refer some examples of works that used the calibrated antecedent rainfall model, the reference Zêzere et al (2015) did not used this method as you can verify in the original source. It was used in a previous work by "Zêzere and Rodrigues (2002) Rainfall thresholds for landsliding in Lisbon Area (Portugal). In: Rybar J, Stemberk J, Wagner P(eds) Landslides. Lisse, A. A. Balkema, pp 333–338".*

**Response:** Many thanks for the valuable recommendation. The literature review was corrected and the related reference was added.

***10.*** *Section 3.1 – general climate conditions – second paragraph – "the long term climatological shows. . .." – please refer how many years correspond to this long period of climatological data.*

**Response:** Here we added the related reference.

Page 6, line 7: "Chub (2007) confirms that long-term climatology based on 50 stations data (some of the stations have more than 100 years historical records) in Uzbekistan shows…."

***11.*** *Page 6, line 25 – "The spatial distribution of the average precipitation. . ." – again please refer the period of this data.*

**Response:** Period data was referred to the Figure 4 (1961-1990). However, in revised manuscript we removed the figure due to the cartographical issues that we couldn't improve and we added the related reference (Chub, 2007) as it was consistent with the figure results.

***12.*** *Section 3.2 – mudflows in Uzbekistan – you should specify which are the data sources of the archive data of mudflows.*

**Response:** We have added the data source (Uzhydromet) on page 7, line 6: "The archive data of mudflow occurrences in Uzbekistan has been collected by Uzhydromet since then."

***13.*** *Page 7 line 26 – why do you use the terms episodes and events of mudflows? It has the same meaning? Please clarify this topic in the text.*

**Response:** We proposed to mean mudflow occurrences by using the terms episodes and events in page 7 line 26. Yes, it has the same meaning.

*14. Authors relate the highest peaks of mudflows in the 1930s, 1960s and 1990s. I ask if these peaks can be related with some land use changes occurred in the country.*

**Response**: We agree that this is relevant and important question that requires further research. However, according to Chub (2007), apart from the natural causes the number of recorded mudflows can be increased due to the several factors which often interact to generate mudflow occurrences in Uzbekistan. Social-economic factors such as residential and industrial activities below unstable hillslopes accelerate soil creep as well as accumulated materials in channels decrease the roughness condition by overloading with fills which can potentially increase the probability and impact of mudflows. We expanded the discussion according.

*15. Sections 4 and 5 are more related to the results presentation, so the text needs to be improved and better structured. In the section 4.1 the text must be limited to the results and the part related to the methods can be included in section 2.*

**Response:** We agree. We removed the first paragraph from 4.1 and we included the text in section 2 Methodology part of the manuscript and improved the text.

***Notes about the figures: This work should reduce the number of figures and tables.***

**Response:** We followed this recommendation and reduced the number of figures in revised manuscript.

*1. Figure 1 – it misses a scale and a north arrow in the map, and also the name of the neighboring countries.*

**Response***:* We summarised both reviewers' feedbacks on Figure 1 and produced a new graph presented below.

[Figure]

**Figure 1.** Mudflow occurrences for the years 2005-2014 in areas with high probability of mudflow passage in Uzbekistan: Zerafshan Basin (blue dots) in central part of the country; Fergana Valley (red dots) in the east; Chirchik-Akhangaran Basin (orange) in the north-east; Surkhandarya (green) and Kashkadarya (violet) rivers' basins in the south of Uzbekistan. Map also

represents political administrative divisions and administrative centers/cities of the country. Map does not include inland water resources.

*2. Figure 2 – figure caption is confusing. It can be only methodological flowchart.*

**Response:** We have changed Figure 2 caption from "Schematic diagram of pathways by which the stages of investigation presented in this paper" to "Methodological flowchart of the investigation process presented in this paper"

*3. Figure 3 - these stations are located in which basin?*

**Response:** We added information regarding the location of the stations in each basin respectively. We also changed the colour to increase visibility of the figure.

[Figure]

Figure 3. The 30-year means (1984-2013) of monthly temperature (°C, red line) and precipitation (mm, green bars) in four selected stations namely Gallyaral in Zerafshan Basin (a), Chimgan in Chirchik-Akhangaran Basin (b), Mingchukur representing Kashkadarya and Surkhandarya Basins (c) and Sokh in Fergana Valley (d) with high occurrences of mudflow in Uzbekistan. Graphs have different scales.

*4. Figure 4 – this map has several cartographical problems. The legend and the north arrow should be reduced because its size is over exaggerated in comparison to the map size. Erase the title "distribution of precipitation" because the figure caption will detail that information. Again it misses the scale in this map. This figure caption can be simply "total annual precipitation. . ..." because there is no need to repeat that this is a map.*

**Response:** We would like to improve this figure, however, it was adapted from the presentation material

https://www.unece.org/fileadmin/DAM/env/water/meetings/Assessment/Almaty%20workshop/pdf/day1/Agaltseva_UZ_Climate_Change.pdf in Russian by Natalya Agaltseva (Uzhydromet) on "The impact of climate change on water resources in Uzbekistan" (page 7, bottom left graph). Reviewer 1 has suggested to remove the Figure 4, and we agree with him as long as we cannot improve the map due to the data access used in this graph.

**5.** *Figure 6 – the figure caption can be only "Monthly mudflow frequencies. . ."*

**Response:** Many thanks. Done.

**6.** *Figure 7 – it misses the legend of the codes used in this figure in order to be understandable.*

**Response***:* Figure 7 was presented as originally published. However, due to the first reviewer and your kind suggestions we produced a figure based on the reference material and added information about the nature of the flow.

[Figure]

**Figure 7.** Scheme of synoptic weather types in Central Asia and Uzbekistan during the cold (a) and warm seasons (b) of the year (after Inagamova et al., 2002). Blue and red cursors indicate relatively cold and warm air trajectories approaching to the investigation area (grey background). Abbreviations and numbers of each weather type explained in Table 1 of Appendices.

**7.** *Figure 8 – please specify for which area corresponds this graph.*

**Response:** Generally, this figure relates to the whole study area, precisely for the whole country. Figure 8 was produced for illustrative purposes only.

**8.** *Figure 10 – b) it misses the legend of the triangles or put the name of the stations in the figure. C) include the units of the legend (Fig10c) and remove the blue from the legend because it has no cartographical representation.*

**Response:** Many thanks for the comment. First anonymous referee has also suggested minor improvements in Figure 10. We summarised both reviewers' suggestions and we improved the figure as presented below.

[Figure]

**Figure 10. a)** Location of the study domain together with the 16 grid points and central grid point (40N-67.5E, red circle) used in the automated weather circulation type; **b)** Investigation area shown in rectangle and location of selected stations (black triangles) around the central grid point 40.0 N-67.5 E (red circle) of CWT objective method. Stations: Gallyaral (40.02 N-67.60 E), Chimgan (41.57 N-70.00 E), Sokh (39.97 N-71.13 E) and Mingchukur (38.70N – 66.90 E); **c)** ERA-Interim orography map and the location of central grid point (red circle) together with representative four stations (black triangles).

**9. Figure 11, 12 and A2 and A3** *can be joined in two figures. In the first figure you can put a) and b) for the 4 study areas (one page) and in the second one you can put c) and d) for the 4 study areas (one page). With this solution you reduce the number of figures and also the reference of these figures in the text will be clearer. Also reduce the figure caption of these figures, removing some text that can be referred in the text.*

**Response***:* Many thanks. Done.

[Figure]

**Figure 11.** Contribution of CWT classes to the observed precipitation over the stations Gallyaral in Zerafshan Basin (a), Chimgan in Chirchik-Akhangaran Basin (b), Mingchukur representing Kashkadarya and Surkhandarya Basins (c) and Sokh in Fergana Valley (d) for warm (Mar-Aug) and cold (Sep-Feb) seasons for the years 1984-2013. CWT days - frequency of each class in percentage; % total precipitation - contribution of each class to the overall precipitation; mm/day - daily average precipitation per CWT. The figure has different scales.

[Figure]

**Figure 12.** Box plots show daily precipitation (1984-2013) per CWT class in four representative stations namely Gallyaral (Zerafshan Basin), Chimgan (Chirchik-Akhangaran Basin), Mingchukur (Kashkadarya and Surkhandarya Basins) and Sokh (Fergana Valley). The blue and red lines represent 0.90th and 0.95th percentiles of the precipitation for each class. The graph has different scales.

**10. Figure 14** – *reduce the length of the figure caption because it is too long and include the example in the text, not in the caption.*

**Response:** Many thanks for the comment and for suggesting this improvement. We revised Figure14 by adding more precisely figure caption and we also proposed to make a minor correction to the figure.

[Figure]

**Figure 14.** Anomaly of mudflow days per CWT class (grey bars, grey axis, %) and CWT classes for mudflow days (red line, red axis, %) for the March-August period between 1984 and 2013 in five regions: a) Zerafshan Basin (101 days); b) Fergana Valley (147 days); c) Chirchik-Akhangaran Rivers Basin (57 days); d) Kashkadarya Basin (35 days); e) Surkhandarya Basin (44 days). Figure has different scales.

Explanation of the Figure 14 will be added in the text:

"Red line is CWT for mudflow days: 1018 days from total 5520 climatological CWT days in warm season associated with cyclonic circulation over Zerafshan Basin and only 25 days or 2.5 % from total cyclonic days (red line and red scale) resulted mudflow occurrences in Zerafshan (Figure 14a). Similarly mudflow anomalies or values compared to the average occurrence of mudflows (1.8% or 101 days out of 5520) in this region increased nearly 35% on cyclonic days (grey bar and grey scales). However, frequency of anticyclonic days shows 1.3 % (13 anticyclonic mudflow days out of 1012 anticyclonic totals) and decreases more than 20% compared to the average mudflow occurrences. During the investigation period E, SE and S classifications highly unlikely resulted mudflow events in Zerafshan subsequently representing missing values in figure."

**11. Figure 15** – *include the meaning of the red lines in the graphs.*

**Response:** Thank you for the comment. We have added short description of the red lines in figure caption: "Figure 15. Antecedent Daily Rainfall Model applied to the representative stations Gallyaaral (a), Chimgan (b), Mingchukur (c) and Sokh (d) for the period 1984-2013. Red lines and curves indicate 0.1th, 0.5th and 0.9th probability threshold triggering mudflow occurrences in respective regions. Equations from Table 4 used for calculation of the probability lines in selected stations"

**12. Figure 16, 17, A5 and A6** *in my opinion are not necessary to understand the results. Please consider remove these figures or other alternative to reduce the number of figures.*

**Response:** We have joined all these figures into one figure.

[Figure]

**Figure 16.** Threshold probabilities initiating mudflow occurrences per CWT class in the stations namely Gallyaaral, Chimgan, Mingchukur and Sokh (panel columns) for the period of March-April 1984- 2013. Black dot is day without mudflow, green triangle – days with probable mudflow, red triangle is a day initiated mudflow occurrences in the study area. Red lines and curves indicate 0.1th, 0.5th and 0.9th probability threshold triggering mudflow occurrences per CWT class.

**Notes about the tables: 1. Table 1** – it is necessary to put the unit of each row of the table.

**Response***:* Many thanks. Done.

Table 1. Mudflow disasters causing fatalities and other relative damages over the period 2005-2014 in Uzbekistan (Data source: Uzhydromet)

| Year | Number of deaths | Number of households damages | Livestock head counts | Infrastructure damages | | | | | Agricultural crops | | | |
|---|---|---|---|---|---|---|---|---|---|---|---|---|
| | | | | Highways (km) | Local bridges (count) | Hydrologic bridges or tools (count) | Schools (count) | Other (count) | Cotton fields (ha) | Wheat fields (ha) | Gardens (ha) | Other (ha) |
| 2005 | | 860 | | | 1 | | | 2 | 200 | 69 | | |
| 2006 | 7 | 175 | | | | | | 2 | 152 | 165 | 118 | 22 |
| 2007 | | 8 | 1 | 6 | 15 | 7 | | 3 | | 2 | | 6 |
| 2008 | 7 | 413 | 1 | 0.3 | 5 | | | 49 | 747 | 261 | | 123 |
| 2009 | 8 | 498 | 80 | | 14 | 5 | 2 | | 966 | 834 | 56 | 18 |
| 2010 | 8 | 41 | | | 6 | | 2 | 7 | | 5 | | 3 |
| 2011 | 2 | 94 | 50 | 0.5 | | 1 | | 52 | 483.5 | 318.6 | 0.12 | 10.1 |
| 2012 | 5 | 773 | 3 | 2.7 | 25 | 6 | 1 | 55 | | | | |
| 2013 | 1 | 31 | | 0.012 | 2 | 6 | | 3 | | | | 200 |
| 2014 | | | | | | | | 4 | | | | |
| Total | 38 | 2893 | 135 | 10 | 68 | 25 | 5 | 177 | 2548 | 1655 | 174 | 382 |

**References**

CHUB, V. Y. 2007. *Climate change impacts on hydrometeorological processes, agro climatic and water resources of the Republic of Uzbekistan,* Tashkent, Voris Nashriyot, 133p. (in Russian).

INAGAMOVA, S. I., MUKHTAROV, T. M. & MUKHTAROV, S. T. 2002. *General features of synoptic processes of Central Asia. ,* Tashkent, Central Asian Hydrometeorological Scientific Research Institute, 476 p. (in Russian).

---

## Author Response (AR1)

Dear Editor,

We would like to thank you and the reviewers for the insightful comments and feedback on our manuscript, which we have fully addressed them in revised version of the manuscript. To the extent possible, red (strikethroughs) and blue (underlined) fonts have been used to reflect the changes made for 'tracked-change' version of the manuscript. We look forward to hearing from you with any last minor changes in due course.

We hope that the revised manuscript fulfils the requirements for publication in Natural Hazards and Earth System Sciences (NHESS) journal.

On behalf of all authors with kind regards,
Gavkhar Mamadjanova

[revised manuscript text omitted]

**Commented [GM(wISAS+2]:** We checked all other acronyms and hope all are explained now

**Commented [GM(wISAS+3]:** We have gone through the paper and included reference to figures as Figure

The desired outcome of this study is to eventually select representative weather types which can then be applied to AOGCM and RCM. That way the influence of precipitation patterns on mudflow occurrences can be studied under climate change scenarios across Uzbekistan and CA in further studies.

**2.1 Data**

5   The investigation is based upon two categories of datasets: ground observation and reanalysis. Observed daily meteorological variables recorded by Uzhydromet, such as precipitation and temperature from the four meteorological stations (Gallyaral, Sokh, Chimgan and Mingchukur) located in the mountains and the foothills  with high mudflow passages were used to produce respective climatologies (Table 2).

In addition, historical data of Uzhydromet regarding mudflow occurrences in Uzbekistan over the period 1870-2014 were
10   analysed.  National scale mudflow database includes information such as the name of the water stream, location, date of passage, the potential reason for the formation of the mudflow,  a rough estimate of the volume and major damages.

Data of the daily mean synoptic situation or local classification of Synoptic Weather Types (SWT), which is available at Library Services and the Archive Department of Uzhydromet as  catalogues of six hourly (00, 06, 12, 18 GMT) data manually derived from synoptic charts, was calculated to produce relative outputs
15   regarding mudflow inducing weather situations. The period considered in this study is a warm season (March-August) of 1984-2013.

In order to assess potential climatic drivers over Uzbekistan daily mean lower atmospheric flow in 700 hPa geopotential height (GPH) fields by  the European Centre for Medium-Range Weather Forecasts (ECMWF) ERA-Interim reanalysis (Dee et al., 2011), spanning the period 1984-2013, was used to estimate the large-scale atmospheric circulation. This gridded data
20   set has 0.75° spatial resolution.

**2.2 Methods**

**2.2.1 Stations and Mudflow Events Selection**

We select stations namely Gallyaaral (574 m a.s.l.) located in Zerafshan Basin, Sokh (1200 m a.s.l.) in Fergana Valley,
25   Chimgan (1620 m a.s.l.) in Chirchik-Akhangaran Basin and Mingchukur (2132 m a.s.l.) representing both Kashkadarya and Surkhandarya Basins (Figure 9b-9c, Table 2) in order to investigate regional and local characteristics of the rainfall over each basin with high mudflow passages and its adjustment areas. The investigation area is limited to rain gauges located in the mountain areas, based on the assumption that each station well represents hydro-climatologic conditions of the basin and at the same time station data on rainfall accumulation process can capture mudflows within the radius of roughly 100 km.
30   Regional and local mudflow data was extracted from the national scale database for the warm season (March-August) of 1984-2013 to evaluate the relationship between mudflows and synoptic and large scale meteorological patterns. There were more

Commented [GM(wISAS+4]: We added a new table in order to give characteristics of the station and station data used in this study

Commented [GM(wISAS+5]: Subsection 2.2.1 was added due to the Referee#2 comments regarding mudflows selection in the study area

[revised manuscript text omitted]

Commented [GM(wISAS+11):** We removed this figure and produced a new graph presented below

5

**Commented [GM(wISAS+12]:** We summarised both reviewers' feedbacks on Figure 1 and produced a new graph

[Figure]

**Figure 1.** Mudflow occurrences for the years 2005-2014 in areas with high probability of mudflow passage in Uzbekistan: Zerafshan Basin (blue dots) in central part of the country; Fergana Valley (red dots) in the east; Chirchik-Akhangaran Basin (orange) in the north-east; Surkhandarya (green) and Kashkadarya (violet) rivers' basins in the south of Uzbekistan. Map also represents political administrative divisions and administrative centers/cities of the country.

[Figure]

**Figure 2.**  Methodological flowchart of the investigation process presented in this paper.

[Figure]

Commented [GM(wISAS+13]: We have changed the colour to increase visibility of the figure presented below.

**Figure**  **5.** The 30-year means (1984-2013) of monthly temperature (°C,  red line) and precipitation (mm,  green bars) in four selected stations namely (Gallyaaral in Zerafshan Basin (a), Chimgan in Chirchik-Akhangaran Basin (b), Mingchukur representing Kashkadarya and Surkhandarya Basins (c) and Sokh  in Fergana Valley (d) with high occurrences of mudflow in Uzbekistan. Graphs have different scales.

Commented [GM(wISAS+14]: We added information regarding the location of the stations in each basin respectively

[Figure]

**Figure 4.** Map showing the spatial distribution of total annual precipitation (mm) in Uzbekistan for the 1961-1990 periods (Source: Uzhydromet).

Commented [GM(wISAS+15]:** We followed Referee#1 suggestion and removed Figure 4 from the manuscript

[Figure]

**Figure 5 6**. Variability of mudflow events in Uzbekistan (1870-2014). Vertical bars present the mudflow observations for each year. The mean annual mudflow count (21) is indicated in a solid continues horizontal line (pink). Curves (red, blue, green) have been fitted to the distribution for illustrative purposes, denote the 5, 11 and 21-year rates of mudflow occurrences.

[Figure]

**Figure 6 7.** Monthly mudflow frequencies (bars) for the years 1870-2014. Values over the bars indicate the percentage of mudflow occurrences in a given month.

**Commented [GM(wISAS+16]:** Figure caption has been corrected

[Figure]

**Commented [GM(wISAS+17]:** We produced a new figure given below based on the reference material and added information about the nature of the flow.

**Figure  3.** Scheme of synoptic weather types in Central Asia and Uzbekistan during the cold (a) and warm seasons (b) of the year (after Inagamova et al., 2002). . Blue and red cursors indicate relatively cold and warm air trajectories approaching to the investigation area (grey background). Abbreviations and numbers of each weather type explained in Table 2 of Appendices.

[Figure]

Commented [GM(wISAS+18): We added abbreviations in order to indicate the respective weather class

[Figure]

**Figure 8 4.** Frequency distributions of daily synoptic weather types by Bugayev's classification during the cold (Sep-Feb) and warm (Mar-Aug) seasons in the period of 1935-2014 y. Definitions of SWT can be seen in Table 1 2 in appendices section.

[Figure]

Commented [GM(wISAS+19):] We added abbreviations for the figure in order to indicate the respective weather class

[Figure]

[Figure]

**Figure  8.** Frequency of mudflows under the synoptic weather types (SWT) over Uzbekistan during 1984-2013 (March-August): a) Zerafshan Basin (101 days); b) Fergana Valley (147 days); c) Chirchik-Akhangaran Rivers Basin (57 days); d) Kashkadarya Basin (35 days); e) Surkhandarya Basin (44 days). Definitions of SWT can be seen in Table  2 in appendices section.

[Figure]

**Commented [GM(wISAS+20]:** Grey background in Figures 9a and 9b was changed to a blue colour to indicate inland water resources. Investigation area was filled in grey background. Orography unit (m) in Figure 9c was added. New produced figure given below.

**Figure  9. a)** Location of the study domain together with the 16 grid points and central grid point (40N-67.5E, red circle) used in the automated weather circulation type; **b)** Investigation area shown in rectangle and location of selected stations (black triangles)  around the central grid point 40.0 N-67.5 E (red circle) of CWT objective method. Stations : Gallyaaral (40.02 N-67.60 E), Chimgan (41.57 N-70.00 E), Sokh (39.97 N-71.13 E) and Mingchukur (38.70N – 66.90 E); **c)** ERA-Interim orography map and the location of central grid point (red circle) together with representative  four stations (black triangles).

[Figure]

Commented [GM(wISAS+21]: Following Referee#2 recommendation we joined **Figure 11, 12 and A2 and A3** in two figures. In the first figure we put bar plots (a and b) for the 4 study areas as one page (Figure 10) and in the second graph (Figure 11) we include box plots (c and d) for the 4 study areas respectively.

**Figure 11.** Contribution of CWT classes to the observed precipitation over Gallyaral station (Zerafshan Basin), for warm (a) and cold (b) seasons during the period 1984-2013. CWT days - frequency of each class in percentage; % total precipitation - contribution of each class to the overall precipitation; mm/day - daily average precipitation per CWT. Box plots (c, d) show the statistical analysis of daily precipitation per CWT class. The black lines in boxes represent medians for each weather type; the lower (upper) box limits mean the first (third) quartiles; the lower (upper) whiskers show the minimum and maximum values of the precipitation; blue and red lines represent $0.90^{th}$ and $0.95^{th}$ percentiles of the precipitation. The figure has different scales.

[Figure]

**Figure 11 10.** Contribution of CWT classes to the observed precipitation over  Gallyaaral  in (Zerafshan Basin (a), Chimgan in Chirchik-Akhangaran Basin (b), Mingchukur representing Kashkadarya and Surkhandarya Basins (c) and Sokh in Fergana Valley (d) for warm (a) (Mar-Aug) and cold (b) (Sep-Feb) seasons  for the years 1984-2013. CWT days - frequency of each class in percentage; % total precipitation - contribution of each class to the overall precipitation; mm/day - daily average precipitation per CWT.  The figure has different scales.

.

[Figure]

**Figure 12.** As Figure 11, but for Sokh station (Fergana Valley).

Commented [GM(wISAS+22]: Following Referee#2 recommendation we joined **Figure 11, 12 and A2 and A3** in two figures. In the first figure we put bar plots (a and b) for the 4 study areas as one page (Figure 10) and in the second graph (Figure 11) we include box plots (c and d) for the 4 study areas respectively.

[Figure]

**Figure 11.** Box plots show daily precipitation (1984-2013) per CWT class in four representative stations namely Gallyaaral (Zerafshan Basin), Chimgan (Chirchik-Akhangaran Basin), Mingchukur (Kashkadarya and Surkhandarya Basins) and Sokh (Fergana Valley). The blue and red lines represent 0.90th and 0.95th percentiles of the precipitation for each class. The graph has different scales.

f)

[Figure]

**Figure**  **12.** Frequency of CWT (700GPH) climatology for the period March-August, 1984-2013 (red bars) and mudflow days (blue bars) occurred in Zerafshan Basin (a), Fergana Valley (b), Chirchik-Akhangaran (c), Kashkadarya (d) and Surkhandarya (e) basins in Uzbekistan. Central grid point is  40.0 N - 67.5 E.

[Figure]

Commented [GM(wISAS+23]: We revised the figure adding more precisely figure caption and we also proposed to make minor correction to the figure

Figure  **13**.  Anomaly of mudflow days per CWT class (grey bars, grey axis, %) and CWT classes for mudflow days (red line, red axis, %)  for the March-August period between 1984 and 2013  in five regions: a) Zerafshan Basin (101 days); b) Fergana Valley (147 days); c) Chirchik-Akhangaran Rivers Basin (57 days); d) Kashkadarya Basin (35 days); e) Surkhandarya Basin (44 days). Figure has different scales. ~~(Zerafshan, Fergana, Chirchik-Akhangaran, Kashkadarya and Surkhandarya) and in contrast to the average number of mudflow days per CWT (grey bars, %). For example, of the total 5520 CWT days in warm season throughout 30 years (Mar-Aug, 1984-2013), there were up to 1018 days of cyclonic circulation with the frequency of 2.5 % (25 days) which associated with mudflow events in Zerafshan Valley. Furthermore, it was identified that mudflow events occurred on cyclonic days increased up to 34% in comparison to the average value of mudflow days (1.8% or 101 mudflow days against 5520 CWT days) there.~~

[Figure]

**Figure  14.** Antecedent Daily Rainfall Model applied to the representative stations Gallyaaral (a), Chimgan (b), Mingchukur (c) and Sokh (d) for the period 1984-2013. Red lines and curves indicate 0.1th, 0.5th and 0.9th probability threshold triggering mudflow occurrences in respective regions. Equations from Table 4 used for calculation of the probability lines in selected stations.

Commented [GM(wISAS+24]: We included the meaning of the red lines in the graphs

[Figure]

Commented [GM(wISAS+25]: We have joined all figures (Figure 16, 17, A4 and A5) into one panel plot presented in Figure 15

[Figure]

**Figure  15.** Threshold probabilities initiating mudflow occurrences per CWT class in the stations namely Gallyaaral , Chimgan, Mingchukur and Sokh (panel columns) for the period of March-April 1984-2013. Black dot is a day without mudflow, green triangle – days with probable mudflow, red triangle is a day initiated mudflow occurrences in the study area. Red lines and curves indicate 0.1[th], 0.5[th] and 0.9[th] probability threshold triggering mudflow occurrences per CWT class.

[Figure]

Figure 17. As Figure 16, but for Sokh station (Fergana Valley).

Commented [GM(wISAS+26]: We have joined all figures (Figure 16, 17, A4 and A5) into one panel plot presented in Figure 15

**Tables**

[revised manuscript text omitted]

**Commented [GM(wISAS+31]:** Following Referee#2 recommendation we joined **Figure 11, 12 and A2 and A3** in two figures. In the first figure we put bar plots (a and b) for the 4 study areas as one page (Figure 10) and in the second graph (Figure 11) we include box plots (c and d) for the 4 study areas respectively.

[Figure]

**Commented [GM(wISAS+32]:** Following Referee#2 recommendation we joined **Figure 11, 12 and A2 and A3** in two figures. In the first figure we put bar plots (a and b) for the 4 study areas as one page (Figure 10) and in the second graph (Figure 11) we include box plots (c and d) for the 4 study areas respectively.

**Figure 3.** As Figure 11, but for Mingchukur station (Kashkadarya Basin)

[Figure]

**Commented [GM(wISAS+33]:** We have joined all figures (Figure 16, 17, A4 and A5) into one panel plot presented in Figure 15

**Figure 4.** As Figure 16, but for Chimgan station (Chirchik-Akhangaran Basin)

[Figure]

Commented [GM(wISAS+34]: We have joined all figures (Figure 16, 17, A4 and A5) into one panel plot presented in Figure 15

**Figure 5.** As Figure 16, but for Mingchukur station (Kashkadarya Basin)